# Semi-Supervised Neural Super-Resolution for Mesh-Based Simulations

**Jiyeon Kim** [1]   **Youngjoon Hong** [2]   **Won-Yong Shin** [1]

## Abstract

Mesh-based simulations provide high-fidelity solutions to partial differential equations (PDEs), but achieving such accuracy typically requires fine meshes, leading to substantial computational overhead. Super-resolution techniques aim to mitigate this cost by reconstructing high-resolution (HR), high-fidelity solutions from low-cost, low-resolution (LR) counterparts. However, training neural networks for super-resolution often demands large amounts of expensive HR supervision data. To address this challenge, we propose SuperMeshNet, an HR-data-efficient super-resolution framework for mesh-based simulations aided by message passing neural networks (MPNNs). At its core, SuperMeshNet introduces **complementary learning**, a semi-supervised approach that effectively leverages both 1) a small amount of paired LR–HR data and 2) abundant unpaired LR data via two jointly trained, complementary MPNN-based models. Additionally, our model is enriched by **inductive biases**, which are empirically shown to further improve super-resolution performance. Extensive experiments demonstrate that SuperMeshNet requires 90% less HR data to achieve even lower root mean square error (RMSE) than that of the fully supervised benchmark without the inductive biases. The source code and datasets are available at https://github.com/jykim-git/SuperMeshNet.git.

## 1. Introduction

Mesh-based simulations—such as the finite element method (FEM), or finite volume method (FVM)—are widely used to obtain high-fidelity solutions to partial differential equations (PDEs) across a range of scientific and engineering domains. In mesh-based simulations, the mesh size is carefully chosen to balance computational costs against solution fidelity: finer meshes offer higher fidelity but incur significantly greater computational expenses (Obiols-Sales et al., 2024). Super-resolution techniques are developed to alleviate this trade-off by predicting high-resolution (HR) simulation results from low-resolution (LR) counterparts, thereby aiming to deliver high-fidelity solutions at a reduced cost (Barwey et al., 2024; Obiols-Sales et al., 2024). However, training super-resolution models via conventional fully supervised learning demands substantial quantities of computationally expensive HR data, making data collection a significant bottleneck (Obiols-Sales et al., 2024). In this context, improving the HR data efficiency of super-resolution model training is of paramount importance in reality.

As summarized in Table 1, several unsupervised learning approaches have tackled this challenge but have their own limitations. For example, PhySRNet (Arora, 2022) performs super-resolution without any HR data by incorporating PDEs and constraints into its loss function; however, PhySRNet uses a finite-difference scheme for derivative calculations, which limits its applicability to irregular meshes. MAgNet (Boussif et al., 2022) offers an alternative with zero-shot super-resolution through an interpolator trained on LR data; yet, the prediction error of MAgNet is much larger than that of supervised methods (see Appendix I.1). To the best of our knowledge, *semi-supervised learning* has never been applied to the super-resolution task for mesh-based simulations. This may be partly due to the limited exploration of semi-supervised regression methods, especially those compatible with message passing neural networks (MPNNs), compared to semi-supervised classification.

On one hand, many related studies (Yonekura et al., 2023; Arora, 2022; Li & McComb, 2022; Obiols-Sales et al., 2024) on super-resolution for mesh-based simulations rely heavily on convolutional neural networks (CNNs), which cannot directly handle irregular mesh structures. CNNs require interpolating irregular mesh-based data onto a regular grid, which often necessitates a significantly larger number of nodes to achieve the same fidelity as an irregular mesh, leading to relatively lower computational efficiency. It is worth noting that while regular-grid settings in fluid dynamics can

[1]School of Mathematics and Computing (Computational Science and Engineering), Yonsei University, Seoul, Republic of Korea [2]Department of Mathematical Sciences, Seoul National University, Seoul, Republic of Korea. Correspondence to: Won-Yong Shin <wy.shin@yonsei.ac.kr>.

*Proceedings of the $43^{rd}$ International Conference on Machine Learning*, Seoul, South Korea. PMLR 306, 2026. Copyright 2026 by the author(s).

*Table 1.* Comparison between prior studies and our work. Here, $r = \frac{\text{number of HR data samples}}{\text{number of LR data samples}}$.

| Reference | Learning method | Model |
|---|---|---|
| (Li & McComb, 2022), (Yonekura et al., 2023), (Obiols-Sales et al., 2024) | Fully supervised ($r = 1$) | CNN |
| (de Avila Belbute-Peres et al., 2020), (Barwey et al., 2024) | Fully supervised ($r = 1$) | MPNN |
| (Arora, 2022) | Unsupervised ($r = 0$) | CNN |
| (Boussif et al., 2022) | Unsupervised ($r = 0$) | MPNN |
| SuperMeshNet (ours) | **Semi-supervised** ($0 < r \ll 1$) **(complementary)** | MPNN **(inductive biases)** |

lead to significant LR–HR discrepancies, our primary focus is on discrepancies induced by *complex geometries*, which are poorly handled by conventional CNNs. On the other hand, some studies have adopted MPNNs, such as graph convolutional networks (GCNs) (de Avila Belbute-Peres et al., 2020) or SRGNN (Barwey et al., 2024), which can directly handle irregular mesh data. However, the design and impact of inductive biases on enhancing mesh-based super-resolution performance remain largely underexplored.

To address these limitations, we propose SuperMeshNet, an HR data-efficient super-resolution framework tailored for mesh-based simulations under *very scarce HR supervision*, which differs from the supervised and unsupervised settings. This is the first general framework that can be applied across diverse MPNN architectures for the super-resolution task. Specifically, SuperMeshNet introduces two key components: **complementary learning** and **inductive biases for MPNNs.** First, the **complementary learning** is a *semi-supervised learning* method that exploits a small amount of paired LR–HR data for supervised learning, while leveraging a large pool of unpaired LR data in an unsupervised manner. Our complementary learning is built upon two models; an MPNN-based primary model predicts HR solutions from LR counterparts, while an MPNN-based auxiliary model predicts the ***difference*** between two HR solutions corresponding to two LR counterparts. The predictions from each model are utilized to calculate *pseudo-ground truths*, which serve as the ground truth for the other, enabling *mutual supervision*. Since conventional semi-supervised methods typically employ two identical models (Tarvainen & Valpola, 2017; Dai et al., 2023), they often produce highly similar pseudo-ground truths, making them less informative. On the other hand, owing to distinct but interrelated input–output configurations of our complementary learning, the auxiliary model is capable of capturing intra-resolution relationships while the primary model focuses on inter-resolution relationships. This division of roles fosters synergies in mutual supervision, enhancing super-resolution performance while reducing training time compared to prior semi-supervised strategies.

Second, to further improve the performance of mesh-based super-resolution, we introduce **inductive biases for MPNNs**, guided by our empirical observation. Specifically, we employ two MPNN-architecture-agnostic inductive biases: **node-level centering** and **message-level centering**. The node-level centering centers each node embedding by subtracting the global mean of all node embeddings from each node embedding, while the message-level centering performs a similar centering operation over aggregated messages.

We carry out extensive experiments to validate the effectiveness of our two components in SuperMeshNet. Our results demonstrate that, even with only a small portion (*e.g.,* 10%) of paired LR–HR data, SuperMeshNet surpasses a fully supervised (*e.g.,* 100% paired) benchmark method lacking inductive biases in terms of the root mean square error (RMSE). We also prove that the injected inductive biases consistently reduce the RMSE across six different MPNN architectures, underscoring their general applicability. Finally, our main contributions are summarized as follows:

• **MPNN-agnostic applicability.** SuperMeshNet provides a general super-resolution framework for mesh-based simulations, applicable to various MPNNs, under very scarce HR supervision scenarios.

• **Complementary learning.** To the best of our knowledge, this is the first attempt to incorporate semi-supervised learning compatible with MPNNs into super-resolution for mesh-based simulations.

• **Inductive biases.** We introduce node-level centering and message-level centering, which can substantially enhance super-resolution performance across different MPNN types.

## 2. Methodology

### 2.1. Problem Definition

Briefly, we aim to predict an HR solution $\hat{u}_h$ from an LR solution $u_l$ of the same PDE, while relying on as few HR solutions $u_h$ as possible for training. Formally, let $\Omega \subset \mathbb{R}^D$ be the computational domain on which the PDE is solved. Here, $D$ denotes the spatial dimension. A parameter $\mu$

represents all variations of PDE instances, such as material coefficients, domain geometry, or boundary conditions. For example, $\mu$ could be an angle of an applied force or an aspect ratio of an elliptical hole (see Figures 13–15 in Appendix G). Each choice of $\mu$ defines a different PDE instance. As depicted in Figure 1, we discretize $\Omega$ with an LR mesh $M_l = (P_l, E_l)$ and an HR mesh $M_h = (P_h, E_h)$, where $P_l \in \mathbb{R}^{n_l \times D}$ and $P_h \in \mathbb{R}^{n_h \times D}$ are the positions of the nodes on $M_l$ and $M_h$, respectively, and $E_l$ and $E_h$ are edges on $M_l$ and $M_h$, respectively. Running a PDE solver on these meshes yields LR and HR solutions, which we regard as an LR data sample $u_l$ and an HR data sample $u_h$, defined on the nodes on $M_l$ and $M_h$, respectively. Our objective is to predict $\hat{u}_h \in \mathbb{R}^{n_h \times d}$ from $u_l$ as closely as possible to $u_h$, while minimizing the amount of HR data $u_h$ required for training, where $d$ denotes the dimension of the solution field.

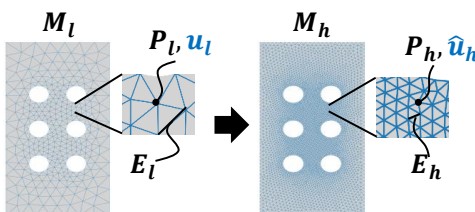

*Figure 1.* Problem setting. We aim to predict $\hat{u}_h$ on HR mesh $M_h$, containing nodes at positions $P_h$ and edges $E_h$, from LR data sample $u_l$ defined on LR mesh $M_l$, comprising nodes at positions $P_l$ and edges $E_l$.

## 2.2. Complementary Learning[1]

### 2.2.1. DATASET SETTING

As depicted in Figure 2, complementary learning in Super-MeshNet leverages both a paired LR–HR training dataset $\mathcal{D}_a = \{(u_l^q, u_h^q) \mid q = 1, 2, \cdots, N_h\}$ containing $N_h$ LR–HR data pairs $(u_l^q, u_h^q)$ and an unpaired LR training dataset $\mathcal{D}_b = \{u_l^q \mid q = N_h + 1, N_h + 2, \cdots, N\}$ having $N - N_h$ LR data samples. The total number of LR data samples is $N$, among which only $N_h$ have HR counterparts, with $N_h \ll N$ in practice. In other words, $N - N_h$ fewer HR data samples are required compared to fully supervised learning. Here, the superscript $q$ is simply an index to distinguish different samples corresponding to different parameters $\mu$. For instance, if $\mu$ is the angle of an applied force, then $(u_l^1, u_h^1)$ corresponds to one angle $\mu^1$, and $(u_l^2, u_h^2)$ corresponds to another $\mu^2$.

### 2.2.2. THE TWO MODELS: $F_\theta$ AND $G_\phi$

To fully exploit unpaired LR data, our complementary learning framework leverages mutual supervision between two models, a primary model $F_\theta$ and an auxiliary model $G_\phi$, which are jointly trained with distinct roles. Unlike con-

---

[1]Pseudo-code is presented in Appendix C.

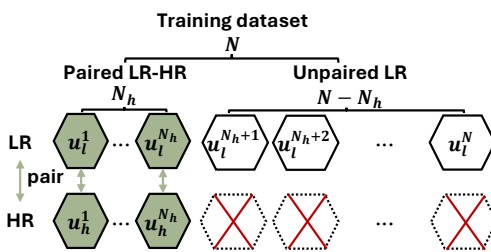

*Figure 2.* Dataset setting. Complementary learning utilizes a paired LR–HR training dataset, including $N_h$ paired data samples (green hexagons), and an unpaired LR training dataset, containing $N - N_h$ unpaired LR data samples (white hexagons). In total, complementary learning can reduce $N - N_h$ HR data samples compared to the case of fully supervised learning.

ventional semi-supervised methods, where multiple models generate pseudo-labels for the same prediction target—often resulting in correlated errors and confirmation bias—our framework decomposes learning into two structurally distinct tasks: inter-resolution mapping (LR → HR) and intra-resolution difference modeling (LR–LR → HR–HR). Because the two models predict different targets, their errors are inherently less correlated than in conventional co-training (see Appendices B and I.7.2). Consequently, the auxiliary model $G_\phi$ provides complementary supervisory signals that improve the overall effectiveness of the framework. From a physical perspective, the two HR solutions are governed by the same PDE and differ only in a parameter $\mu$. Their difference therefore captures the system's physical response to parameter variation. Accordingly, $G_\phi$ learns the solution's sensitivity to parameter perturbations and guides $F_\theta$ to produce physically consistent predictions under varying conditions.

The primary model $F_\theta$, which is used for inference only, predicts an HR solution $\hat{u}_h^q$ from its LR counterpart $u_l^q$:

$$F_\theta(u_l^q) = \hat{u}_h^q \quad (\text{ground truth} : u_h^q). \tag{1}$$

On the other hand, the auxiliary model $G_\phi$, which is used only during training, predicts the ***difference between two HR*** solutions $\hat{u}_h^{rs}$ corresponding to two LR input samples $u_l^r, u_l^s$ to further utilize intra-resolution relations. Here, $r$ and $s$ indicate two LR samples corresponding to different values of parameter $\mu$. Since computational geometry may vary across samples with different $\mu$, HR samples $u_h^r$ and $u_h^s$ may be defined on different positions $P_h^r$ and $P_h^s$. Thus, direct subtraction is not possible. To resolve this, we apply $k$-nearest neighbor ($k$NN) interpolation (Qi et al., 2017) to project solutions defined on $P_h^s$ onto $P_h^r$:

$$G_\phi(u_l^r, u_l^s) = \hat{u}_h^{rs}$$
$$(\text{ground truth} : u_h^r - kNN(u_h^s; P_h^s \to P_h^r)). \tag{2}$$

In $k$NN interpolation, each target point is assigned a distance-weighted average of its $k$ nearest source points (see Appendix D for the details).

**1. Sample paired LR-HR training data** ⬡ **& unpaired LR training data** ⬡

**2. Make predictions with** $F_\theta$ ⬡ **&** $G_\varphi$ ⬡

| Paired LR-HR training data |
| Unpaired LR training data |
| Prediction by $F_\theta$ |
| Prediction by $G_\varphi$ |

**3. Update model parameters of** $F_\theta$ **&** $G_\varphi$ **using total loss** $\mathcal{L} = \mathcal{L}_{F,sup} + \mathcal{L}_{G,sup} + \mathcal{L}_{F,unsup} + \mathcal{L}_{G,unsup}$

*Figure 3.* A schematic overview of complementary learning in SuperMeshNet. It first samples paired LR–HR data $(u_l^\alpha, u_h^\alpha)$, $(u_l^\beta, u_h^\beta)$ and unpaired LR data $u_l^\gamma$. Complementary learning leverages both supervised and unsupervised learning to jointly train two neural network models, $F_\theta$ and $G_\phi$. $F_\theta$ predicts an HR solution from its LR counterpart, while $G_\phi$ predicts the difference between two HR solutions from two LR counterparts to enable synergistic mutual supervision. More specifically, for supervised learning, $F_\theta$ and $G_\phi$ are trained with pairs of LR–HR data (green hexagons). In unsupervised learning, the prediction of one model (yellow and purple hexagons predicted by $F_\theta$ and $G_\phi$, respectively) is used to calculate a pseudo-ground truth that serves as the target for training another model (as depicted by solid and dotted red arrows). The model parameters of $F_\theta$ and $G_\phi$ are updated using the total loss, including both supervised and unsupervised losses based on predictions from both $F_\theta$ and $G_\phi$.

### 2.2.3. LEARNING PROCEDURE

As depicted in Figure 3, each training step combines supervised and unsupervised learning of the two models $F_\theta$ and $G_\phi$. In other words, the loss functions for training $F_\theta$ and $G_\phi$, denoted by $\mathcal{L}_F$ and $\mathcal{L}_G$, respectively, are expressed as:

$$\mathcal{L}_F = \mathcal{L}_{F,sup} + \mathcal{L}_{F,unsup} \quad (3)$$

$$\mathcal{L}_G = \mathcal{L}_{G,sup} + \mathcal{L}_{G,unsup}, \quad (4)$$

where the subscripts $sup$ and $unsup$ represent supervised and unsupervised learning, respectively. To this end, three samples are randomly sampled: 1) two paired LR samples $u_l^\alpha$ and $u_l^\beta$ from the paired LR–HR dataset $\mathcal{D}_a$ for supervised learning and 2) one additional unpaired LR sample $u_l^\gamma$ from the unpaired LR dataset $\mathcal{D}_b$ for unsupervised learning. Here, $\alpha$, $\beta$, and $\gamma$ are the indices referring to distinct parameters $\mu$.

**In supervised learning**, an HR data sample $u_h^\alpha$ is available. As depicted in Figure 3, $F_\theta$ is trained to reduce the MSE between its prediction $\hat{u}_h^\alpha$ and its target, which is the ground truth $u_h^\alpha$. An analogous procedure is applied to $\beta$, thus resulting in:

$$\mathcal{L}_{F,sup} = \ell(\hat{u}_h^\alpha, u_h^\alpha) + \ell(\hat{u}_h^\beta, u_h^\beta), \quad (5)$$

where $\ell(\cdot, \cdot)$ denotes the MSE. Similarly, as expressed in Eq. (2), $G_\phi$ is trained to reduce the MSE between its predictions $\hat{u}_h^{\alpha\beta}$ and its target, which is the ground truth difference between $u_h^\alpha$ and $u_h^\beta$ using the following loss:

$$\mathcal{L}_{G,sup} = \ell(\hat{u}_h^{\alpha\beta}, u_h^\alpha - kNN(u_h^\beta; P_h^\beta \to P_h^\alpha)). \quad (6)$$

**In unsupervised learning**, the ground truth HR data sample $u_h^\gamma$ is unavailable. Under this circumstance, we leverage *mutual supervision* between $F_\theta$ and $G_\phi$. For example, as depicted in Figure 3, if $G_\phi(u_l^\gamma, u_l^\alpha)$ predicts $\hat{u}_h^{\gamma\alpha}$ that approximates the difference between two HR samples $u_h^\gamma - u_h^\alpha$, then adding this to the known $u_h^\alpha$ yields an estimate of $u_h^\gamma$. This pseudo-ground truth can serve as a target for $F_\theta(u_l^\gamma)$. Similarly, if $F_\theta(u_l^\gamma)$ produces $\hat{u}_h^\gamma$ close to $u_h^\gamma$, then subtracting known $u_h^\alpha$ from $\hat{u}_h^\gamma$ provides an approximation of $u_h^\gamma - u_h^\alpha$, which can serve as a target for $G_\phi(u_l^\gamma, u_l^\alpha)$, accordingly. An analogous procedure is applied to the pair $(\beta, \gamma)$. It should be noted that a thorough treatment of $kNN$ interpolation is also required to effectively handle mesh mismatches. For example, in Eq. (7), $u_h^\beta - \hat{u}_h^{\beta\gamma}$ can be used to approximate $u_h^\gamma$. However, the values are defined on position $P_h^\beta$, whereas the prediction $\hat{u}_h^\gamma$ is defined on position $P_h^\gamma$. To reconcile this discrepancy, $kNN$ interpolation is employed to project the values from $P_h^\gamma$ onto $P_h^\beta$. The resultant loss functions are:

$$\mathcal{L}_{F,unsup} = \ell(\hat{u}_h^\gamma, \hat{u}_h^{\gamma\alpha} + kNN(u_h^\alpha; P_h^\alpha \to P_h^\gamma))$$
$$+ \ell(\hat{u}_h^\gamma, kNN(u_h^\beta - \hat{u}_h^{\beta\gamma}; P_h^\beta \to P_h^\gamma)) \quad (7)$$

$$\mathcal{L}_{G,unsup} = \ell(\hat{u}_h^{\gamma\alpha}, \hat{u}_h^\gamma - kNN(u_h^\alpha; P_h^\alpha \to P_h^\gamma))$$
$$+ \ell(\hat{u}_h^{\beta\gamma}, u_h^\beta - kNN(\hat{u}_h^\gamma; P_h^\gamma \to P_h^\beta)). \quad (8)$$

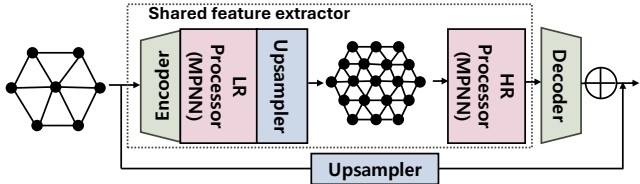

Figure 4. Model architecture of $F_\theta$.

Figure 5. Model architecture of $G_\phi$.

## 2.3. Model Architecture

The architecture of the primary model $F_\theta$, illustrated in Figure 4, is built upon SRGNN (Barwey et al., 2024). The role of $F_\theta$ is to transform LR data into HR data, which is conducted by the lowermost upsampler in Figure 4. To surpass the performance of $k$NN interpolation by the lowermost upsampler, we introduce additional upsampling in latent space. Specifically, an encoder maps the physical quantities into high-dimensional latent space. The LR processor applies message passing to refine LR representations, which are then upsampled to HR latent embeddings. Subsequently, the HR processor applies additional message passing to further enhance the HR representations. Finally, a decoder maps the latent embeddings back to the physical space. The final HR output is obtained by adding the two upsampled HR fields: one from the $k$NN-based upsampler and the other from the latent-space upsampling pathway.

The auxiliary model $G_\phi$, visualized in Figure 5, is responsible for predicting the difference between two HR samples corresponding to two LR input samples. It extends $F_\theta$ to accommodate two input samples, maintaining a comparable structure. A notable architectural feature is the use of a shared feature extractor between $F_\theta$ and $G_\phi$, which helps reduce computational costs during training. First, latent embeddings are extracted from the two LR inputs using the shared feature extractor. Then, one embedding is subtracted from another and the result is decoded to predict the HR difference. The final HR output is obtained by adding the two upsampled HR fields: one from the $k$NN-based upsampler and the other from the latent-space upsampling pathway. The interpolators in Figure 5 serve only to address mesh mismatches when the two LR samples are defined on different meshes.

## 2.4. Inductive Biases for MPNNs

The key difference between the two models, $F_\theta$ and $G_\phi$, and SRGNN (Barwey et al., 2024) is that the MPNNs in the LR and HR processors are enriched with inductive biases: **node-level centering** and **message-level centering**. We illustrate these inductive biases using an MPNN based on MeshGraphNet (MGN) (Pfaff et al., 2021), which serves as the default MPNN architecture throughout our experiments. Implementation details for various types of MPNNs (including MGN) are provided in Appendix F.

In an MPNN layer, a target node $i$ receives messages $msg_{ij}$ from its neighboring nodes $j \in \mathcal{N}(i)$. In MGN, the message is defined as the edge embedding $e_{ij}$, where $e_{ij}$ is computed from the node embeddings $x_i$ and $x_j$ of nodes $i$ and $j$ using a learnable multilayer perceptron (MLP) $MLP_e$:

$$e_{ij} \leftarrow MLP_e(x_i, x_j, e_{ij}),$$
$$msg_{ij} = e_{ij}, \tag{9}$$

where the exact definition of $msg_{ij}$ depends on the MPNN types.

Each node $i$ then aggregates messages $msg_{ij}$ from its neighbors $j$, and the aggregated message $agg_i$ is used to update its node embedding by employing a learnable MLP $MLP_x$:

$$agg_i = \sum_{j \in \mathcal{N}(i)} msg_{ij},$$
$$x_i \leftarrow MLP_x(x_i, agg_i). \tag{10}$$

We empirically find that injecting the following two inductive biases into the message passing mechanism substantially improves super-resolution performance. The first inductive bias, **node-level centering**, subtracts the mean of all node embeddings $x_i$ from each individual $x_i$ after the node embedding update in Eq. (10):

$$x_i \leftarrow x_i - \frac{1}{n} \sum_{i=1}^{n} x_i, \tag{11}$$

where $n$ denotes the number of nodes in the LR mesh $M_l$ or the HR mesh $M_h$. For MPNNs that explicitly compute aggregated messages $agg_i$, such as MGN, **message-level centering** subtracts the mean of $agg_i$ from each individual aggregated message between the message aggregation and the node embedding update in Eq. (10):

$$agg_i \leftarrow agg_i - \frac{1}{n} \sum_{i=1}^{n} agg_i. \tag{12}$$

The centering operations tend to smooth the loss landscape, similar to the effect reported for batch normalization (Santurkar et al., 2018), which may facilitate optimization. However, because centering removes global mean information, it is beneficial only for tasks that do not rely heavily on such information (e.g., super-resolution). Experimental validations of these arguments are provided in Appendix I.3.

*Table 2.* Summary of FEM datasets.

| Dataset | Equation | Solution | Parameter | LR nodes | HR nodes |
|---------|----------|----------|-----------|----------|----------|
| 1 | Linear elasticity | von Mises stress | Force angle | 333 | 4,053 |
| 2 | Linear elasticity | von Mises stress | Hole shape | 329–387 | 3,959–4,157 |
| 3 | Poisson equation | Electric field | Hole shape | 324–388 | 3,959–4,154 |

*Table 3.* Summary of CFD datasets.

| Dataset | Equation | Solution | Parameter | LR nodes | HR nodes |
|---------|----------|----------|-----------|----------|----------|
| Real-world geometry dataset | Incompressible Navier-Stokes | Velocity, pressure | Angle of attack | 15,619 | 86,087 |
| Time-dependent PDE dataset 1 | Incompressible Navier-Stokes | Velocity, pressure | Time | 5,200 | 20,400 |
| Time-dependent PDE dataset 2 | Incompressible Navier-Stokes | Vorticity | Time | 1,024 | 1,048,576 |

## 3. Experimental Results and Analyses[2]

### 3.1. Datasets[3]

**FEM Datasets.** Table 2 summarizes three FEM datasets used in our experiments, detailing their governing PDEs, the quantities derived from solving these equations, the parameters that vary across the samples, and the number of nodes in LR and HR meshes. Dirichlet boundary conditions are utilized for all datasets. Datasets 2 and 3 include elliptical geometries with varying aspect ratios, leading to various magnitude differences between LR and HR fields despite similar spatial patterns, making these datasets particularly challenging for super-resolution.

**CFD Datasets.** To validate the applicability of SuperMesh-Net to a complex real-world geometry and a time-dependent PDE, we adopt three computational fluid dynamics (CFD) datasets. As summarized in Table 3, the real-world geometry dataset is constructed by solving the incompressible Navier-Stokes equation around a motorbike with a rider. Since our primary focus is on handling LR–HR field discrepancies arising from complex geometries, this real-world benchmark dataset is particularly challenging. The time-dependent PDE dataset 1 is obtained by solving the incompressible Navier–Stokes equations for flow around a cylinder. The time-dependent PDE dataset 2 is generated following (Kochkov et al., 2021a) to construct a dataset in which the HR vorticity differs substantially from the LR vorticity. Meanwhile, for the time-dependent PDE dataset 1, LR data samples are generated by downsampling HR data onto LR meshes because we found that time synchronization between independently simulated LR and HR trajectories is not guaranteed. For all other datasets, LR data samples are obtained by independently solving the governing PDEs on LR meshes.

---

[2]Additional experimental results and analyses are provided in Appendix I.

[3]A detailed description of the datasets is provided in Appendix G.

### 3.2. Experimental Setup

We evaluate our methodology using six representative MPNNs, including GCN (Kipf & Welling, 2017), Graph-SAGE (SAGE) (Hamilton et al., 2017), GAT (Veličković et al., 2018), Graph Transformer (GTR) (Shi et al., 2021), GIN (Xu et al., 2019), and MGN (Pfaff et al., 2021). Each MPNN consists of three layers for LR processing and additional three layers for HR processing. The default hidden dimension is set to 30; however, for larger datasets—specifically, the time-dependent PDE 2 dataset and the real-world geometry dataset—it is increased to 150 and 120, respectively. For datasets that predict both velocity and pressure—namely, the time-dependent PDE dataset 1 and the real-world geometry dataset—we replace the standard MSE loss with a weighted MSE loss to account for the scale difference between velocity and pressure. The weights assigned to velocity and pressure are $99 : 1$ and $10^{-8} : 1$ for the time-dependent PDE dataset 1 and the real-world geometry dataset, respectively. Throughout the experiments, we adopt the Adam optimizer with a learning rate of $1 \times 10^{-3}$ and PyTorch's automatic mixed precision training to improve computational efficiency. All experiments are carried out on a machine with Intel (R) Core (TM) i9-10920X CPUs@3.50 GHz and an NVIDIA RTX A6000 GPU. The RMSE is used as a metric where lower values indicate better performance.

### 3.3. Comparison with Full Supervision

Table 4 compares the RMSE of each MPNN integrated with our framework SuperMeshNet, and its variant without inductive biases, SuperMeshNet-O, against two fully supervised baselines—the same type of MPNNs but trained with full supervision without inductive biases. SuperMesh-Net-O trained with 20 HR data samples (*i.e.*, $N_h = 20$) and 200 LR data samples (*i.e.*, $N = 200$) achieves a significantly lower RMSE compared to the MPNNs trained exclusively on 20 paired LR–HR samples (*i.e.*, $N_h = N = 20$). The improvement is attributed to complementary learning, which is

*Table 4.* The RMSE of SuperMeshNet (with inductive biases) and SuperMeshNet-O (without inductive biases) trained with $N_h = 20$ HR data samples and $N = 200$ LR data samples across six MPNNs and three datasets, in comparison with two fully supervised MPNNs including 1) $N_h = N = 20$ and 2) $N_h = N = 200$. The best performer is highlighted in **bold**. The main advantage of our approach lies in **reducing HR data requirements by 90% ($200 \rightarrow 20$) while maintaining the accuracy of full supervision**, rather than in improving absolute RMSE itself.

| | Method | $N_h$, $N$ | MPNN | | | | | |
|---|---|---|---|---|---|---|---|---|
| | | | GCN | SAGE | GAT | GTR | GIN | MGN |
| Dataset 1 | Fully supervised | 20, 20 | 0.0874 | 0.0876 | 0.0826 | 0.0758 | 0.0819 | 0.0655 |
| | Fully supervised | 200, 200 | 0.0575 | 0.0544 | 0.0512 | 0.0450 | 0.0381 | 0.0228 |
| | SuperMeshNet-O | 20, 200 | 0.0613 | 0.0589 | 0.0544 | 0.0451 | 0.0404 | 0.0269 |
| | SuperMeshNet | 20, 200 | **0.0431** | **0.0450** | **0.0457** | **0.0385** | **0.0277** | **0.0226** |
| Dataset 2 | Fully supervised | 20, 20 | 0.0972 | 0.1025 | 0.0983 | 0.0983 | 0.0775 | 0.0730 |
| | Fully supervised | 200, 200 | 0.0624 | 0.0633 | 0.0637 | **0.0572** | **0.0534** | **0.0461** |
| | SuperMeshNet-O | 20, 200 | 0.0636 | 0.0664 | 0.0680 | 0.0631 | 0.0569 | 0.0514 |
| | SuperMeshNet | 20, 200 | **0.0574** | **0.0624** | **0.0634** | 0.0600 | 0.0537 | 0.0507 |
| Dataset 3 | Fully supervised | 20, 20 | 0.0587 | 0.0611 | 0.0616 | 0.0513 | 0.0569 | 0.0523 |
| | Fully supervised | 200, 200 | 0.0370 | 0.0340 | 0.0374 | 0.0329 | 0.0317 | **0.0243** |
| | SuperMeshNet-O | 20, 200 | 0.0380 | 0.0366 | 0.0375 | 0.0363 | 0.0316 | 0.0281 |
| | SuperMeshNet | 20, 200 | **0.0297** | **0.0297** | **0.0310** | **0.0294** | **0.0258** | 0.0245 |

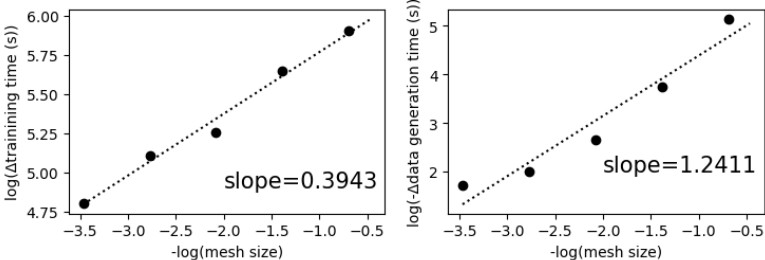

*Figure 6.* Training time increase (left) and data generation time decrease (right), resulting from the use of SuperMeshNet ($N_h = 20$, $N = 200$), relative to fully supervised learning ($N_h = N = 200$) on Dataset 1 and its mesh-size variants. All experiments use MGN as the underlying MPNN architecture.

inherently designed to effectively leverage the 180 unpaired LR samples that fully supervised learning cannot utilize. Remarkably, despite being trained only with 20 HR data samples, SuperMeshNet-O achieves RMSE values that are on par with the second fully supervised baseline trained with the entire 200 HR data samples (*i.e.*, $N_h = N = 200$). SuperMeshNet, enriched by inductive biases, surpasses the second fully supervised baseline ($N_h = N = 200$) in most cases, highlighting the efficacy of the proposed inductive biases tailored for super-resolution in improving performance. This implies the potential to reduce up to 90% of the effort required to generate HR data. It is worth noting that the main advantage of our approach lies in reducing HR data requirements by 90% while maintaining the accuracy of full supervision, rather than in improving absolute RMSE itself. Furthermore, our findings consistently demonstrate the improvement of SuperMeshNet in terms of HR data efficiency across all six MPNNs. This underscores its versatility and effectiveness in enhancing super-resolution performance, regardless of the type of underlying MPNN architecture.

### 3.4. Overall Training Cost

SuperMeshNet alleviates the reliance on expensive HR data samples, reducing training data generation time. However, it incurs longer training time compared to the fully supervised baseline. Figure 6 displays how the training time increase and data generation time decrease—resulting from the use of SuperMeshNet ($N_h = 20$, $N = 200$)—scale with mesh size, relative to fully supervised learning ($N_h = N = 200$) on Dataset 1 (the full set of results on all FEM datasets as well as device settings are available in Appendix I.5.1). Comparison of the two slopes, which characterize how training time increases and data generation time decreases with mesh size, respectively, reveals that data generation time grows more rapidly as mesh size decreases. This trend is particularly important in our target regime, where fine meshes lead to significant computational cost. We expect that, for sufficiently fine meshes, the savings in data generation time will outweigh the additional training time, resulting in an overall reduction in computational costs.

## 3.5. Comparison with Benchmark Semi-Supervised Regression Methods

Table 5 compares complementary learning in SuperMesh-Net against benchmark semi-supervised regression methods. As presented in Table 5, SuperMeshNet achieves the lowest RMSE while also exhibiting the shortest training time among all benchmark semi-supervised regression methods. The performance improvements achieved by SuperMesh-Net likely stem from its inherent characteristics of using two complementary models. Mean-Teacher (Tarvainen & Valpola, 2017) and UCVME (Dai et al., 2023) employ two models to predict the same target, *i.e.*, an HR data sample. Similarly, TNNR (Wetzel et al., 2022) uses one twin neural network to predict the difference between two HR data samples. On the other hand, SuperMeshNet employs the *complementary learning* mechanism that leverages two distinct yet cooperative models: primary model $F_\theta$, which learns to predict an HR data sample, and auxiliary $G_\phi$, which learns to predict the difference between two HR data samples, as formulated in Eq. (2). While $F_\theta$ operates on a single LR input, $G_\phi$ utilizes two LR data samples along with one HR data sample, enabling the two models to make predictions from distinct informational viewpoints. This design reduces the correlation of errors between the two models' predictions, thereby enabling complementary mutual supervision.

*Table 5.* Comparison with benchmark semi-supervised regression methods in terms of the RMSE and training time (in seconds). Here, MGN is employed as an MPNN for each method. Training is conducted when $N_h = 20$ and $N = 200$ for Dataset 1. The best value in each metric is highlighted in **bold**.

| Methods | RMSE | Training time (s) |
|---|---|---|
| Mean-Teacher (Tarvainen & Valpola, 2017) | 0.0325 | 693.84 |
| TNNR (Wetzel et al., 2022) | 0.0624 | 477.48 |
| UCVME (Dai et al., 2023) | 0.0293 | 1122.62 |
| SuperMeshNet-O | 0.0269 | 503.2 |
| SuperMeshNet | **0.0226** | **421** |

## 3.6. Ablation Studies on Inductive Biases

Table 6 presents ablation results on the two inductive biases in SuperMeshNet, demonstrating their effect on super-resolution performance in terms of the RMSE across six different MPNNs[4]. For all MPNNs, the incorporation of node-level centering (N) and message-level centering (M) into the MPNN architecture leads to substantial improvements in super-resolution performance compared to MPNNs without inductive biases (O). We provide further analysis on the underlying factors contributing to the performance improvement in Appendix I.3.

---

[4]Note that, in MPNNs such as GCN and GAT, the message-level centering cannot be employed independently since the message aggregation in Eq. (10) and the node embedding updates are integrated into a single step.

*Table 6.* Ablation studies on inductive biases. The RMSE of SuperMeshNet across six MPNNs under four inductive bias conditions (O: without inductive biases, N: node-level centering, M: message-level centering, and N+M: both node-level and message-level centering operations) trained with $N_h = 20$ and $N = 200$ for Dataset 1 is compared. For each MPNN, the lowest RMSE value among the four inductive bias conditions is highlighted in **bold**.

| MPNN | RMSE | | | |
|---|---|---|---|---|
| | O | N | M | N + M |
| GCN | 0.0613 | **0.0431** | - | - |
| SAGE | 0.0589 | 0.0493 | 0.0528 | **0.0450** |
| GAT | 0.0544 | **0.0457** | - | - |
| GTR | 0.0451 | 0.0405 | 0.0438 | **0.0385** |
| GIN | 0.0404 | 0.0290 | 0.0281 | **0.0277** |
| MGN | 0.0269 | 0.0237 | 0.0247 | **0.0226** |

## 3.7. Application to Real-World Geometry and Time-Dependent PDE

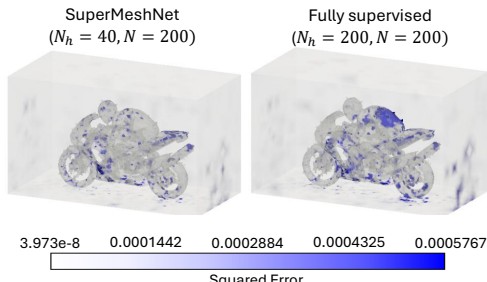

SuperMeshNet
($N_h = 40, N = 200$)  Fully supervised ($N_h = 200, N = 200$)

| 3.973e-8 | 0.0001442 | 0.0002884 | 0.0004325 | 0.0005767 |

Squared Error

*Figure 7.* Comparison of the squared error of pressure between SuperMeshNet and fully supervised baselines on a real-world geometry dataset. Here, $N_h$ and $N$ denote the numbers of HR and LR data samples, respectively. For all cases, MGN is used as the underlying MPNN.

*Table 7.* Comparison with the fully supervised baseline in predicting physical quantities (drag and lift coefficients) on the real-world geometry dataset. Here, MGN is used as the MPNN backbone for all methods.

| Methods | Drag coefficient (relative error) | Lift coefficient (relative error) |
|---|---|---|
| Ground truth HR data | 0.3724 | 0.0368 |
| SuperMeshNet ($N_h = 40, N = 200$) | 0.3778 (0.014) | 0.0433 (0.177) |
| Fully supervised ($N_h = N = 200$) | 0.3653 (0.019) | 0.0380 (0.033) |

Figure 7 demonstrates the applicability of SuperMeshNet to real-world geometry when MGN is used as the MPNN backbone. It visualizes the error distribution on the real-world geometry dataset, where darker blue indicates higher error. Notably, SuperMeshNet, trained with only 40 HR samples, achieves even lower errors than the case of a fully supervised model trained with 200 HR samples, particularly around the *back of the rider*. These results indicate that SuperMeshNet remains effective in handling complex geometric structures.

*Table 8.* Comparison with the fully supervised baseline in predicting physical quantities (amplitudes of drag and lift coefficients) on the time-dependent PDE dataset 1. Here, MGN is used as the MPNN backbone for all methods.

| Methods | Amplitude of drag coefficient (relative error) | Amplitude of lift coefficient (relative error) |
|---|---|---|
| Ground truth HR data | 0.0451 | 0.6984 |
| SuperMeshNet ($N_h = 40, N = 200$) | 0.0441 (0.0232) | 0.6907 (0.0111) |
| Fully supervised ($N_h = N = 200$) | 0.0435 (0.0360) | 0.6568 (0.0595) |

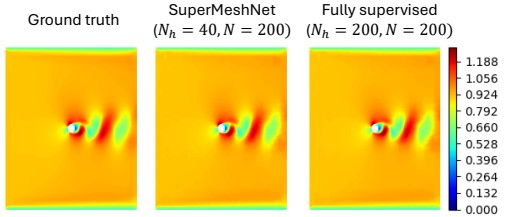

*Figure 8.* Comparison of the ground truth and predicted fluid flow speeds on the time-dependent PDE dataset 1. Here, $N_h$ and $N$ represent the number of HR and LR data samples, respectively. For all cases, MGN is utilized as the underlying MPNN.

*Figure 9.* Comparison of the LR input, ground truth HR data, and two vorticity predictions produced by SuperMeshNet and full supervision on the time-dependent PDE dataset 2. Here, $N_h$ and $N$ represent the number of HR and LR data samples, respectively. For all cases, MGN is utilized as the underlying MPNN.

To further validate the superiority of SuperMeshNet from both physical and practical perspectives, we evaluate the drag and lift coefficients, where the drag coefficient quantifies the resistance force acting on the motorbike opposing the flow direction, and the lift coefficient measures the force perpendicular to it. Table 7 shows that SuperMeshNet achieves a lower relative error than the case of the fully supervised baseline in predicting the drag coefficient, despite using significantly fewer HR samples. In contrast, SuperMeshNet exhibits a slightly larger relative error for the lift coefficient, which can be attributed to its small magnitude—small absolute deviations can lead to large relative errors. Since the absolute error is less than 0.01 for both coefficients, these results indicate that SuperMeshNet remains effective for deriving meaningful physical quantities.

Next, we evaluate the applicability of SuperMeshNet to time-dependent PDEs. Figure 8 presents qualitative comparisons on the time-dependent PDE dataset 1. The predictions of SuperMeshNet, trained with only 40 HR samples, closely match the ground truth and are comparable to those of the fully supervised model trained with 200 HR samples. Furthermore, we evaluate the amplitudes of the drag and lift coefficients of the cylinder, which characterize the wake flow behind it. Table 8 shows that SuperMeshNet predicts these amplitudes with an absolute error below 0.01 and a relative error under 3%, while outperforming the fully supervised baseline despite using fewer HR samples. In addition, Figure 9 presents qualitative comparisons on the time-dependent PDE dataset 2. SuperMeshNet successfully predicts the HR counterpart even when the ground truth differs substantially from the LR input, whereas conventional fully supervised learning fails despite having access to significantly more HR data samples. This advantage stems from our auxiliary model $G_\phi$, which leverages HR–HR relationships during training, while fully supervised baselines rely solely on LR–HR mappings that are more challenging to learn due to the large discrepancies between LR and HR fields.

## 4. Conclusions and Limitations

In this paper, we address the open problem of super-resolution for mesh-based simulations by proposing SuperMeshNet, which leverages complementary learning and inductive biases to achieve high HR data efficiency. Complementary learning enables the effective use of unpaired LR data, while the inductive biases further improve performance across a range of MPNN architectures. While SuperMeshNet substantially reduces the reliance on HR data, several challenges remain for making complementary learning more efficient, effective, and better understood. Complementary learning is slower than fully supervised training. We expect that, for sufficiently fine meshes, the reduction in HR data generation cost will outweigh this additional training overhead; however, further improving the computational efficiency of complementary learning remains an important direction for future work. In addition, while Appendix I.13 provides a preliminary empirical investigation of training stability, a rigorous theoretical characterization is needed to understand when complementary learning remains stable without mutual error amplification. Moreover, Appendix I.12 shows that HR data selection strategies can significantly influence performance, suggesting the need for a more principled understanding of how HR samples should be chosen. Finally, because the auxiliary model learns solution variations under parameter perturbations, the framework may be less effective in regimes with strong nonlinearities or bifurcations. Understanding these failure modes is an important step toward extending SuperMeshNet to more complex physical systems.

## Acknowledgments

The work of W.-Y. Shin was supported by the National Research Foundation of Korea (NRF), South Korea grant funded by the Korea government (MSIT) (RS-2021-NR059723), and by SMEs Technology Innovation Development Program through the Technology Innovation and Promotion Agency (TIPA), funded by Ministry of SMEs and Startups (RS-2024-00511332). The work of Y. Hong was supported by Basic Science Research Program through the NRF funded by the Korea government (MSIT) (RS-2023-00219980), and by Institute of Information & communications Technology Planning & Evaluation (IITP) grant funded by the Korea government (MSIT) [NO.RS-2021-II211343, Artificial Intelligence Graduate School Program (Seoul National University)].

## Impact Statement

This paper introduces SuperMeshNet, a super-resolution framework designed to advance the field of machine learning-aided physical simulations. By providing a cost-effective alternative to traditional simulation methods, SuperMeshNet substantially reduces computational and resource expenses, making high-fidelity simulations more accessible. This efficiency can facilitate the broader adoption of simulations across various engineering disciplines including, but not limited to, solid mechanics, enabling more extensive exploration and optimization while minimizing the need for costly trial-and-error experiments. Ultimately, SuperMeshNet has the potential to accelerate innovation by lowering the barriers to conducting complex simulations.

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

# A. Notations

Table 9 summarizes the notations used throughout the paper.

*Table 9.* Summary of notations.

| Notation | Description |
|---|---|
| $\mu$ | PDE parameter |
| $F_\theta, G_\phi$ | neural network models |
| $u_l, u_l^q, u_l^r, u_l^s, u_l^\alpha, u_l^\beta, u_l^\gamma$ | LR data samples |
| $u_h, u_h^q, u_h^r, u_h^s, u_h^\alpha, u_h^\beta, u_h^\gamma$ | HR data samples |
| $\hat{u}_h, \hat{u}_h^\alpha, \hat{u}_h^\beta, \hat{u}_h^\gamma$ | prediction by $F_\theta$ |
| $\hat{u}_h^{\alpha\beta}, \hat{u}_h^{\beta\gamma}, \hat{u}_h^{\gamma\alpha}$ | prediction by $G_\phi$ |
| $M_l$ | LR mesh |
| $M_h$ | HR mesh |
| $P_l$ | nodal positions of the LR mesh |
| $P_h, P_h^\alpha, P_h^\beta, P_h^\gamma$ | nodal positions of the HR mesh |
| $E_l$ | edges of the LR mesh |
| $E_h$ | edges of the HR mesh |
| $n_l$ | number of nodes in the LR mesh |
| $n_h$ | number of nodes in the HR mesh |
| $n$ | number of nodes in the underlying graph |
| $\mathcal{D}_a$ | paired LR–HR training dataset |
| $\mathcal{D}_b$ | unpaired LR training dataset |
| $N$ | total number of data samples 
     = number of LR data samples |
| $N_h$ | number of paired LR–HR data samples 
     = number of HR data samples |
| $\mathcal{L}_F$ | loss function for training $F_\theta$ |
| $\mathcal{L}_G$ | loss function for training $G_\phi$ |
| $\ell$ | mean squared error |
| $k\mathrm{NN}$ | $k$-nearest neighbor interpolation |
| $x_i$ | node embedding of node $i$ |
| $x_j$ | node embedding of node $j$ |
| $e_{ij}$ | edge embedding between nodes $i$ and $j$ |
| $msg_{ij}$ | message between nodes $i$ and $j$ |
| $agg_i$ | aggregated message of node $i$ |
| $f_m$ | message function |
| $f_x$ | node embedding update function |
| $\mathcal{N}(i)$ | set of neighboring nodes of $i$ |

# B. Related Work

**Super-resolution for simulations.** Similar to surrogate models for simulations, early super-resolution models for simulations predominantly employed CNN-based image super-resolution architectures, such as SRGAN (Li & McComb, 2022) and UNet (Yonekura et al., 2023). As a pioneering work, CFD-GCN (de Avila Belbute-Peres et al., 2020) introduced GCNs for the super-resolution of computational fluid dynamics (CFD) simulations. This method demonstrated both improved generalization to unseen data and enhanced cost efficiency. More recently, advanced MPNN architectures such as SRGNN were applied to the super-resolution of fluid flows (Barwey et al., 2024). Despite these advancements, inductive biases tailored for MPNNs in the context of super-resolution for mesh-based simulations are largely underexplored.

**Semi-supervised regression.** Semi-supervised regression involves predicting real-valued output using both labeled and unlabeled datasets. Compared to semi-supervised classification, semi-supervised regression remains largely underexplored (Kostopoulos et al., 2018). A co-training approach typically splits input features into groups, with each group used to train a separate model (Brefeld et al., 2006). However, in scenarios with limited features, such as FEM-relevant data including only nodal positions and nodal values, splitting features can lead to insufficient information for accurate predictions. As an alternative, CoREG (Zhou & Li, 2005) was presented to eliminate the need for feature splitting by using two $k$-nearest neighbor ($k$NN) regressors with different distance metrics. Unfortunately, this approach is restricted to $k$NN regressors and is unsuitable for predicting values at the node level. A recent method, Rankup (Huang et al., 2024), reformulated regression tasks into classification tasks to leverage a rich set of methodologies developed for semi-supervised classification. However, this technique lacks generalizability for regression tasks involving mesh-based graph data. The Mean-Teacher framework (Tarvainen & Valpola, 2017), though not originally designed for mesh-based data, has potential for dealing with mesh-based graph data. It involves teacher and student models with the teacher's weights updated as the exponential moving average of the student's weights. In contrast, the recently proposed UCVME framework (Dai et al., 2023) has demonstrated superior performance over Mean-Teacher (Tarvainen & Valpola, 2017) by incorporating uncertainty consistency and utilizing a variational model ensemble. However, because Mean-Teacher (Tarvainen & Valpola, 2017) and UCVME (Dai et al., 2023) both employ identically structured models predicting the same target, they exhibit reduced pseudo-label diversity (correlated errors) during training. This uniformity (correlation) diminishes synergy between the models and consequently hinders learning efficiency. Additionally, Twin Neural Network Regression (TNNR) (Wetzel et al., 2022) is applicable to mesh-based predictions when an appropriate model architecture is involved. However, it involves only a single twin neural network, thus lacking the synergistic benefits of mutual supervision.

## C. Pseudo-code for Complementary Learning

We present pseudo-code of our complementary learning mechanism in SuperMeshNet.

---

**Algorithm 1** Complementary learning

---

**Input**: paired LR–HR dataset: $\mathcal{D}_a = \{(u_l^q, u_h^q) \mid q = 1, 2, \cdots, N_h\}$, unpaired LR dataset: $\mathcal{D}_b = \{u_l^q \mid q = N_{h+1}, N_{h+2}, \cdots, N\}$, neural network models: feature extractor $E_c$, $F_\theta$'s decoder $D_F$, and $G_\phi$'s decoder $D_G$, maximum number of epochs: $ep$, learning rate: $\eta$, early stopping criterion

**Output:** Trained neural network models: $E_c$, $D_F$, and $D_G$

**for** $epoch \leftarrow 1$ to $ep$ **do**

  **for** $step \leftarrow 1$ to $N$ **do**

    Sample paired LR–HR data $(u_l^\alpha, u_h^\alpha), (u_l^\beta, u_h^\beta) \in \mathcal{D}_a$

    Sample unpaired LR data $u_l^\gamma \in \mathcal{D}_b$

    Compute node embeddings by $E_c$:

$$x^\alpha \leftarrow E_c(u_l^\alpha), \ x^\beta \leftarrow E_c(u_l^\beta), \ x^\gamma \leftarrow E_c(u_l^\gamma)$$

    Compute prediction by $D_F$:

$$\hat{u}_h^\alpha \leftarrow D_F(x^\alpha), \ \hat{u}_h^\beta \leftarrow D_F(x^\beta), \ \hat{u}_h^\gamma \leftarrow D_F(x^\gamma)$$

    Compute prediction by $D_G$:

$$\hat{u}_h^{\alpha\beta} \leftarrow D_G(x^\alpha, x^\beta), \ \hat{u}_h^{\beta\gamma} \leftarrow D_G(x^\beta, x^\gamma), \ \hat{u}_h^{\gamma\alpha} \leftarrow D_G(x^\gamma, x^\alpha)$$

    Compute loss for $F_\theta$:

$$\begin{aligned}
\mathcal{L}_F =& \ell(\hat{u}_h^\alpha, u_h^\alpha) + \ell(\hat{u}_h^\beta, u_h^\beta) + \ell\big(\hat{u}_h^\gamma, \hat{u}_h^{\gamma\alpha} + kNN(u_h^\alpha; P_h^\alpha \to P_h^\gamma)\big) \\
& + \ell\big(\hat{u}_h^\gamma, kNN(u_h^\beta - \hat{u}_h^{\beta\gamma}; P_h^\beta \to P_h^\gamma)\big)
\end{aligned}$$

    Compute loss for $G_\phi$:

$$\begin{aligned}
\mathcal{L}_G =& \ell\big(\hat{u}_h^{\alpha\beta}, u_h^\alpha - kNN(u_h^\beta; P_h^\beta \to P_h^\alpha)\big) + \ell\big(\hat{u}_h^{\gamma\alpha}, \hat{u}_h^\gamma - kNN(u_h^\alpha; P_h^\alpha \to P_h^\gamma)\big) \\
& + \ell\big(\hat{u}_h^{\beta\gamma}, u_h^\beta - kNN(\hat{u}_h^\gamma; P_h^\gamma \to P_h^\beta)\big)
\end{aligned}$$

    Compute total loss:

$$\mathcal{L} = \mathcal{L}_F + \mathcal{L}_G$$

    Compute gradients: $\nabla_\psi \mathcal{L}$ where $\psi$ is the parameters of $E_c$, $D_F$, and $D_G$

    Update weights:

$$\psi \leftarrow \psi - \eta \nabla_\psi \mathcal{L}$$

  **end for**

  **if** early stopping criterion is met **then**

    Break

  **end if**

**end for**

**return** $E_c$, $D_F$, and $D_G$

---

# D. $k$NN Interpolation in **SuperMeshNet**

We provide a brief explanation of the $k$NN interpolation procedure in SuperMeshNet, which projects values defined on nodes of a source mesh onto nodes of a target mesh.

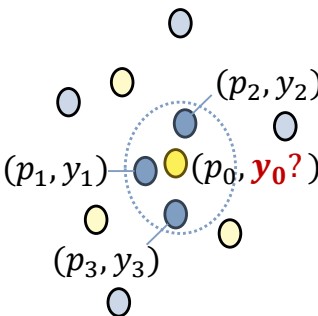

*Figure 10.* Schematic illustration of $k$NN interpolation with $k = 3$. Yellow nodes belong to the target mesh, blue nodes to the source mesh, and the darker blue nodes indicate the $k$ nearest neighbors of the darker yellow node. Given the positions of the $k$ nearest source nodes $p_i$ ($1 \leq i \leq k$), their corresponding values $y_i$, and the target node position $p_0$, the value at the target node $y_0$ can be estimated via weighted averaging.

1. **Find $k$ nearest neighbors ($k$NN).** For each node in the target mesh, identify the $k$ closest nodes in the source mesh. For example, as illustrated in Figure 10, the darker blue nodes represent the three nearest neighbors of the darker yellow node.

2. **Known information.** The nodal positions of the $k$ nearest source nodes $p_i$ ($1 \leq i \leq k$), their values $y_i$, and the target node position $p_0$.

3. **Unknown quantity.** The value at the target node, denoted by $y_0$.

4. **Compute the target node value via weighted averaging.** The interpolation weight for each neighbor is defined as the inverse squared distance from the target node:

$$w_i = \frac{1}{d(p_0, p_i)^2},$$

where $d(\cdot, \cdot)$ denotes the Euclidean distance.

The interpolated value at the target node $y_0$ is then obtained as

$$y_0 = \frac{\sum_{i=1}^{k} w_i y_i}{\sum_{i=1}^{k} w_i}.$$

# E. Model Architectures

The architecture of $F_\theta$, illustrated in Figure 11, is basically built upon SRGNN (Barwey et al., 2024), with the key difference that the MPNNs in the LR and HR processors are enriched with our proposed inductive biases. The $G_\phi$, visualized in Figure 12, extends $F_\theta$ to accommodate two input samples, maintaining a comparable structure. A notable architectural feature is the use of a shared feature extractor between $F_\theta$ and $G_\phi$, which helps reduce computational costs during training. A detailed description of the model architectures follows.

### E.1. Model architecture of $F_\theta$

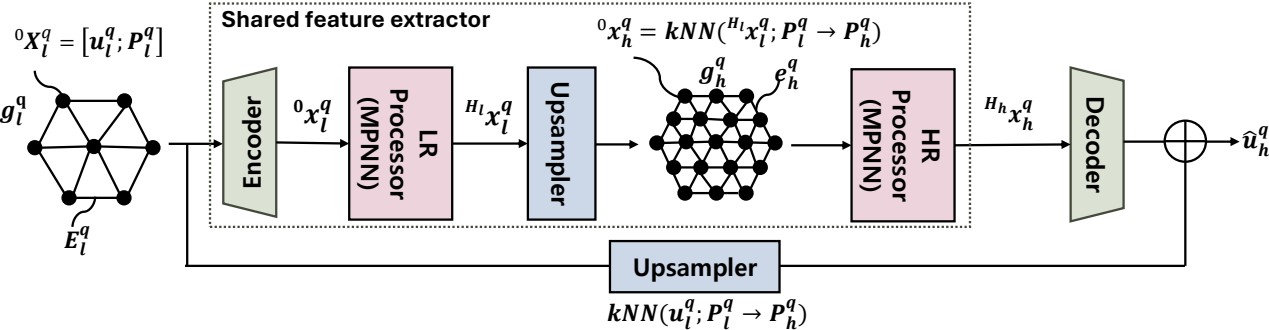

*Figure 11.* The schematic overview of the primary model $F_\theta$. The model $F_\theta$ aims to predict $\hat{u}_h^q$ targeting HR data sample $u_h^q$ from LR data sample $u_l^q$. The LR data sample $u_l^q$ is input to $F_\theta$ as a part of node feature ${}^0X_l^q$ of an input graph $g_l^q$.

The role of $F_\theta$ is to transform LR data into HR data, which is conducted by the lowermost upsampler in Figure 11. To surpass the performance of $k$NN interpolation by the lowermost upsampler, we introduce additional upsampling in latent space. Specifically, an encoder maps the physical quantities into high-dimensional latent space. The LR processor applies message passing to refine LR representations, which are then upsampled to HR latent embeddings. Subsequently, the HR processor applies additional message passing to further enhance the HR representations. Finally, a decoder maps the latent embeddings back to the physical space. The final HR output is obtained by adding the two upsampled HR fields: one from the $k$NN-based upsampler and the other from the latent-space upsampling pathway.

More precisely, $F_\theta$ is designed to make prediction $\hat{u}_h^q$ from an LR data sample $u_l^q$. The LR data sample $u_l^q$ is input to the $F_\theta$ as a form of an input graph $g_l^q$. More specifically, input graph $g_l^q$'s node feature ${}^0X_l^q$ is the concatenation of LR data sample $u_l^q$ and node position $P_l^q$. Depending on MPNN types used in the LR and HR processors (refer to Appendix F), the $g_l^q$ may further include edge feature, which is the concatenation of positions of source and target nodes of the edge $E_l^q$. The $F_\theta$ comprises the encoder, the LR processor, upsamplers, the HR processor, and the decoder. The encoder can be a multi-layer perceptron (MLP) that can convert low-dimensional ${}^0X_l^q$ to high-dimensional node embeddings ${}^0x_l^q$. The LR processor updates the node embedding ${}^0x_l^q$ to ${}^{H_l}x_l^q$ through stacked $H_l$ MPNN layers enriched by our inductive biases. Here, the prescript 0 and $H_l$ indicate the index of the MPNN layers. The node embedding ${}^{H_l}x_l^q$ defined on nodes located at $P_l^q$ is upsampled onto nodes of $g_h^q$ positioned at $P_h^q$ by using $k$NN interpolation. The HR processor similarly updates ${}^0x_h^q$ to ${}^{H_h}x_h^q$ through stacked $H_h$ MPNN layers equipped with our inductive biases. Then, the decoder, which is an MLP, predicts low-dimensional output from ${}^{H_h}x_h^q$. Finally, the LR data sample $u_l^q$ upsampled onto $P_h^q$ by $k$NN interpolation is added to the output of the decoder to obtain the final prediction $\hat{u}_h^q$. The upsampled LR data sample serves as a rough estimation of the prediction, enabling the super-resolution model to focus on learning the finer details, thereby simplifying the learning task.

### E.2. Model architecture of $G_\phi$

The model $G_\phi$ is responsible for predicting the difference between two HR samples corresponding to two LR inputs. Again, to go beyond simple $k$NN-based upsampling by upsampler in Figure 12, we further perform latent-space processing. We extract latent embeddings from the two LR inputs using a shared encoder. The shared encoder is the one used for the model $F_\theta$ depicted in Figure 11. Then, we subtract the embeddings, and decode the result to predict the HR difference. Here, we incorporate subtraction because the goal is to predict the difference between two HR samples. The final HR output is obtained by adding the two upsampled HR fields: one from the $k$NN-based upsampler and the other from the

latent-space upsampling pathway. The interpolators in Figure 12 serve only to address mesh mismatches when the two LR samples are defined on different meshes. Since the underlying computational domain geometry may vary across samples, direct point-wise operations, such as subtraction or addition, are generally infeasible. To overcome this, we apply $k$NN interpolation to project one mesh onto another, enabling consistent alignment between mesh structures.

More precisely, the auxiliary model $G_\phi$ is designed to make prediction $\hat{u}_h^{rs}$ from a pair of LR data samples $u_l^r$ and $u_l^s$. More specifically, the two input LR data samples $u_l^r$ and $u_l^s$ are fed into the $G_\phi$ as parts of node features of two input graphs $g_l^r$ and $g_l^s$, respectively. In order to reduce computational costs, $F_\theta$ and $G_\phi$ share a feature extractor comprising the encoder, the LR processor, the interpolator, and the HR processor. The shared feature extractor returns node embeddings $x_h^r$ and $x_h^s$ from input graphs $g_l^r$ and $g_l^s$, respectively. Then, $x_h^s$ is subtracted from $x_h^s$ to yield $x_h^{rs}$. Here, $k$NN interpolator is used to enable subtraction operation between two node embeddings $x_h^r$ and $x_h^s$ defined at different nodal positions. The $x_h^{rs}$ is fed into the decoder, and the upsampled difference between $u_l^r$ and $u_l^s$ through $k$NN interpolation is added to the decoder's output. Here, the interpolated difference between $u_l^r$ and $u_l^s$ also serves as a rough estimation of the prediction $\hat{u}_h^{rs}$. Again, a $k$NN interpolator is used to enable subtraction operation between $u_l^r$ and $u_l^s$ defined on different nodal positions.

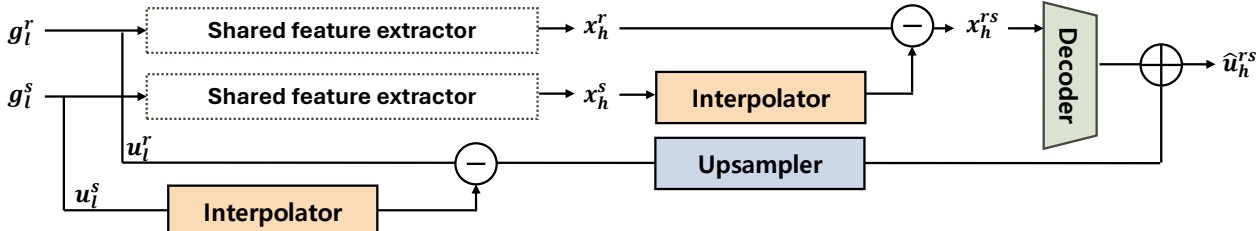

*Figure 12.* The schematic overview of the auxiliary model $G_\phi$. The $G_\phi$ aims to predict $\hat{u}_h^{rs}$ targeting difference between two input LR data samples $u_l^r$ and $u_l^s$. The two input LR data samples $u_l^r$ and $u_l^s$ are fed into the $G_\phi$ as parts of node features of two input graphs $g_l^r$ and $g_l^s$, respectively. In order to reduce computational cost, $F_\theta$ and $G_\phi$ share a feature extractor in Figure 11 consisting of an encoder, an LR processor, an upsampler, and an HR processor.

# F. Incorporation of Inductive Biases into MPNN Architectures

This section describes how inductive biases are incorporated into each of MPNN models.

## F.1. Inductive biases-enriched GCN (Kipf & Welling, 2017)

$$
\begin{aligned}
msg_{ij} &= \Theta^T e_{ji} x_j \\
x_i = agg_i &= \sum_{j \in \mathcal{N}(i)} msg_{ij} \\
x_i &\leftarrow x_i - \frac{1}{n} \sum_{i=1}^{n} x_i,
\end{aligned}
\tag{13}
$$

where $\Theta$ is a learnable parameter and $e_j i$ is an edge weight.

## F.2. Inductive biases-enriched GraphSAGE (SAGE) (Hamilton et al., 2017)

$$
\begin{aligned}
msg_{ij} &= x_j \\
agg_i &= \frac{1}{|\mathcal{N}(i)|} \sum_{j \in \mathcal{N}(i)} msg_{ij} \\
agg_i &\leftarrow agg_i - \frac{1}{n} \sum_{i=1}^{n} agg_i \\
x_i &\leftarrow W_1 x_i + W_2 agg_i \\
x_i &\leftarrow x_i - \frac{1}{n} \sum_{i=1}^{n} x_i,
\end{aligned}
\tag{14}
$$

where $W_1$ and $W_2$ are learnable parameters.

## F.3. Inductive biases-enriched GAT (Veličković et al., 2018)

$$
\begin{aligned}
\alpha_{ij} &= \frac{exp(LeakyReLU(a_s^T \Theta_s x_i + a_t^T \Theta_t x_j))}{\sum_{k \in \mathcal{N}(i)} exp(LeakyReLU(a_s^T \Theta_s x_i + a_t^T \Theta_t x_k))} \\
msg_{ij} &= \alpha_{ij} \Theta_t x_j \\
x_i = agg_i &= \sum_{j \in \mathcal{N}(i)} msg_{ij} \\
x_i &\leftarrow x_i - \frac{1}{n} \sum_{i=1}^{n} x_i,
\end{aligned}
\tag{15}
$$

where $a_s$, $a_t$ $\Theta_s$, and $\Theta_t$ are learnable parameters.

### F.4. Inductive biases-enriched Graph Transformer (GTR) (Shi et al., 2021)

$$
\begin{aligned}
\alpha_{ij} &= softmax((W_3 x_i)^T (W_4 x_j)) \\
msg_{ij} &= \alpha_{ij} W_2 x_j \\
agg_i &= \sum_{j \in \mathcal{N}(i)} msg_{ij} \\
agg_i &\leftarrow agg_i - \frac{1}{n} \sum_{i=1}^{n} agg_i \\
x_i &\leftarrow W_1 x_i + agg_i \\
x_i &\leftarrow x_i - \frac{1}{n} \sum_{i=1}^{n} x_i,
\end{aligned}
\tag{16}
$$

where $W_1$, $W_2$, $W_3$ and $W_4$ are learnable parameters.

### F.5. Inductive biases-enriched GIN (Xu et al., 2019)

$$
\begin{aligned}
msg_{ij} &= x_j \\
agg_i &= \sum_{j \in \mathcal{N}(i)} msg_{ij} \\
agg_i &\leftarrow agg_i - \frac{1}{n} \sum_{i=1}^{n} agg_i \\
x_i &\leftarrow MLP_\Theta((1+\epsilon)x_i + agg_i) \\
x_i &\leftarrow x_i - \frac{1}{n} \sum_{i=1}^{n} x_i,
\end{aligned}
\tag{17}
$$

where $MLP_\Theta$ is a learnable MLP and $\epsilon$ is a learnable parameter.

### F.6. Inductive biases-enriched MeshGraphNet (MGN) (Pfaff et al., 2021)

$$
\begin{aligned}
e_{ij} &\leftarrow MLP_e(x_i, x_j, e_{ij}) \\
msg_{ij} &= e_{ij} \\
agg_i &= \sum_{j \in \mathcal{N}(i)} msg_{ij} \\
agg_i &\leftarrow agg_i - \frac{1}{n} \sum_{i=1}^{n} agg_i \\
x_i &\leftarrow MLP_x(x_i, agg_i) \\
x_i &\leftarrow x_i - \frac{1}{n} \sum_{i=1}^{n} x_i,
\end{aligned}
\tag{18}
$$

where $MLP_e$ and $MLP_x$ are learnable MLPs.

# G. Datasets for Experimental Evaluations

## G.1. Dataset 1

The first dataset is inspired by simustruct (Ribeiro et al., 2023), the dataset for machine learning-based methods in structural analysis. Examples of HR and LR data samples from Dataset 1 are visualized in Figure 13. As depicted in the figure, the computational domain is a rectangle measuring $0.25 \times 0.5$ in the x- and y-directions, containing six circular holes, each with a diameter of 0.05. For the HR mesh, the mesh size around outer four sides is $10 \times 10^{-3}$, while the mesh size around the circular holes is set to be $4 \times 10^{-3}$. For the LR mesh, the mesh size around the outer sides is $40 \times 10^{-3}$, and the mesh size around the circular holes is $16 \times 10^{-3}$.

On the computation domain, the following linear elasticity equation is solved:

$$
\begin{aligned}
-\nabla \cdot \sigma(u) &= 0 \\
\sigma(u) &= \lambda \operatorname{tr}(\epsilon(u))I + 2G\epsilon(u), \\
\epsilon(u) &= \frac{1}{2}\left(\nabla u + (\nabla u)^T\right),
\end{aligned}
\tag{19}
$$

where $\sigma(u)$ is the stress tensor, $\lambda$ and $G$ are Lamé's elasticity parameters for the material, $I$ is the identity tensor, tr is the trace operator on a tensor, $\epsilon(u)$ is the symmetric strain tensor (symmetric gradient), and $u$ is the displacement vector field.

A force of $1 \times 10^8$ is applied to the top side of the rectangle in angles between $40°$ and $140°$ relative to the x-axis, while the bottom side of the rectangle is fixed to zero displacement. Lamé's first and second parameters are 1.25, and $80.8 \times 10^9$, respectively. Von Mises stress is evaluated at each node of the meshes. In order to solve the equation for each dataset, we leverage FEniCSx (Baratta et al., 2023; Scroggs et al., 2022b;a; Alnaes et al., 2014), an open-source computing platform for solving PDEs with the FEM.

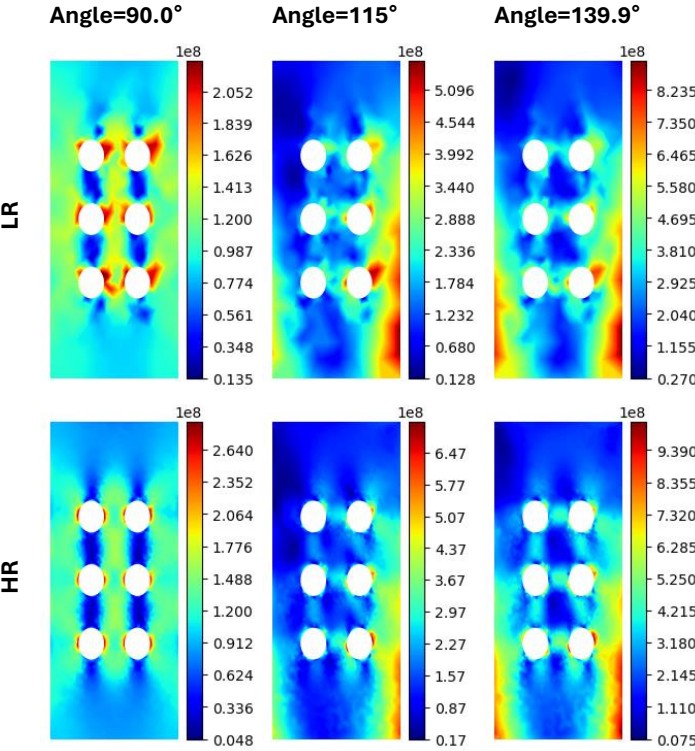

*Figure 13.* Examples of LR and HR data samples with various angles of applied force relative to the x-axis from Dataset 1.

### G.2. Dataset 2

The geometry of the second dataset resembles that of the first dataset, with the primary difference being the shapes of the holes. Specifically, the holes in the second dataset are elliptical, with varying ratios between the lengths of the major and minor axes. The mesh sizes remain the same as those in Dataset 1. Similarly as in Dataset 1, the linear elasticity equation in Eq. (19) is solved. The applied force is directed along the y-axis, while all other conditions and constants remain identical to those in Dataset 1. Examples of HR and LR data samples from Dataset 2 are visualized in Figure 14.

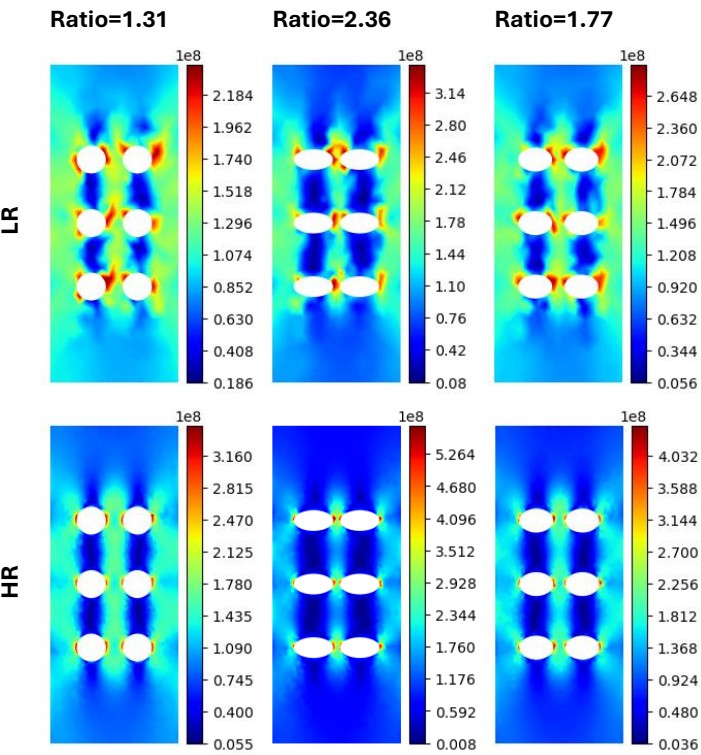

*Figure 14.* Examples of LR and HR data samples with various ratios between the lengths of the major and minor axes from Dataset 2.

### G.3. Dataset 3

The geometry and mesh sizes of Dataset 3 are identical to those of Dataset 2. However, instead of solving the linear elasticity equation, the following Poisson equation is solved.

$$\nabla^2 u = 0, \tag{20}$$

where u is an electrical potential.

The boundary conditions are defined as follows: the four outer sides are set to 0, while the elliptical holes have alternating boundary values. Specifically, the holes centered at (0.08, 0.15), (0.17, 0.25), and (0.08, 0.35) are assigned a value of -1, whereas the holes centered at (0.17, 0.15), (0.08, 0.25), and (0.17, 0.35) are assigned a value of 1. The magnitude of the electric field is calculated at each node of the mesh. Examples of HR and LR data samples from the Dataset 3 are visualized in Figure 15.

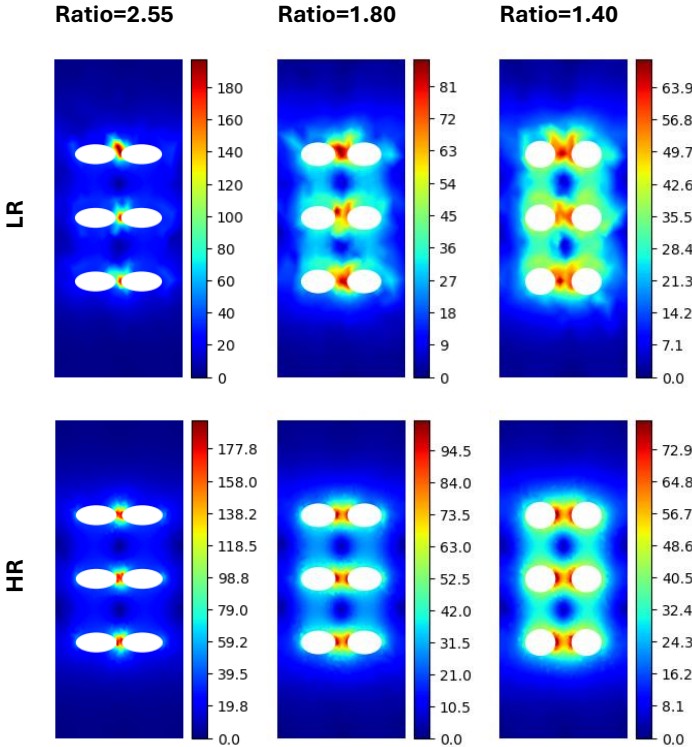

*Figure 15.* Examples of LR and HR data samples with various ratios between the lengths of the major and minor axes from Dataset 3.

## G.4. Real-World Geometry Dataset

An Example of HR and LR data samples from the real-world geometry dataset are visualized in Figure 16. As depicted in the figure, the computational domain is a rectangular box of size $20 \times 8 \times 8$ in the x-, y- and z-directions, containing a rider on a motorbike. The mesh size is set to be finer around the rider and the motorbike. The dataset is built upon a bike tutorial of OpenFOAM (Jasak, 2009) by varying angle of attack from $0°$ to $-90°$. The speed of fluid at the left side of the rectangular box is set to be 20.

To obtain the solution, we use OpenFOAM (Jasak, 2009) in two stages. First, a potential-flow initialization is performed by solving

$$\nabla^2 \phi = 0, \tag{21}$$

and setting

$$u = \nabla \phi, \tag{22}$$

where $u$ denotes velocity and $\phi$ is the velocity potential. This velocity field is used as an initial guess. Then, a steady-state incompressible RANS solver is run to obtain a more accurate velocity and pressure field.

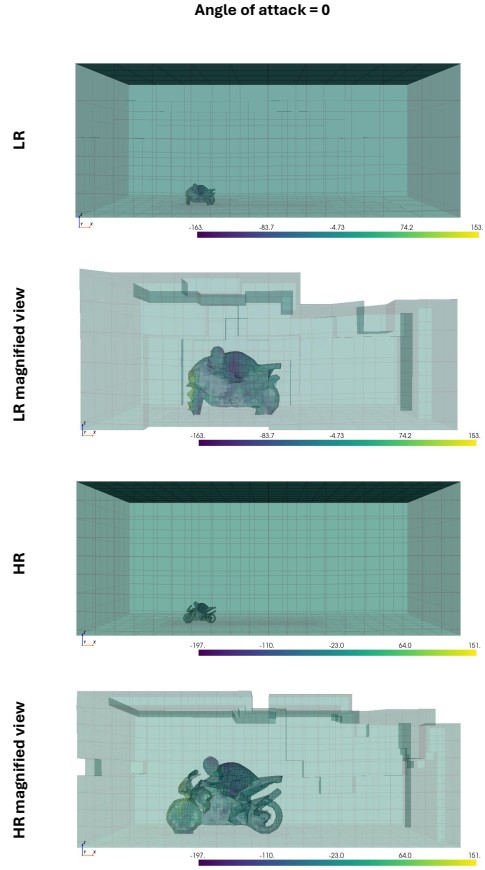

*Figure 16.* An example of LR and HR data samples corresponding to an angle of attack of $0°$ from the real-world geometry dataset.

### G.5. Time-Dependent PDE Dataset 1

Examples of HR and LR data samples from the time-dependent PDE dataset 1 are visualized in Figure 17. As depicted in the figure, the computational domain is a square measuring $2 \times 2$ in the x- and y-directions, containing one cylinder at the center of the domain with a diameter of 0.05. The mesh size is set to be finer around the cylinder.

On the computation domain, the following incompressible Navier-Stokes equation is solved:

$$\nabla \cdot \mathbf{u} = 0, \tag{23}$$

$$\rho \left( \frac{\partial \mathbf{u}}{\partial t} + \mathbf{u} \cdot \nabla \mathbf{u} \right) = -\nabla p + \eta \nabla^2 \mathbf{u}, \tag{24}$$

where $\mathbf{u}$ is velocity, $p$ is pressure, and $\rho$ is density, $\eta$ is dynamic viscosity. The velocity at the left boundary of the square increases from 0 to 1 over the time interval $t \in [0, 0.5]$, and remains constant thereafter. The time step is set to $5 \times 10^{-4}$, and the total simulation time is 40; only data for $t \geq 20$ are used. The density and dynamic viscosity are set to 1 and $5 \times 10^{-4}$, respectively. The governing equations are solved using `OpenFOAM` (Jasak, 2009), an open-source CFD toolbox.

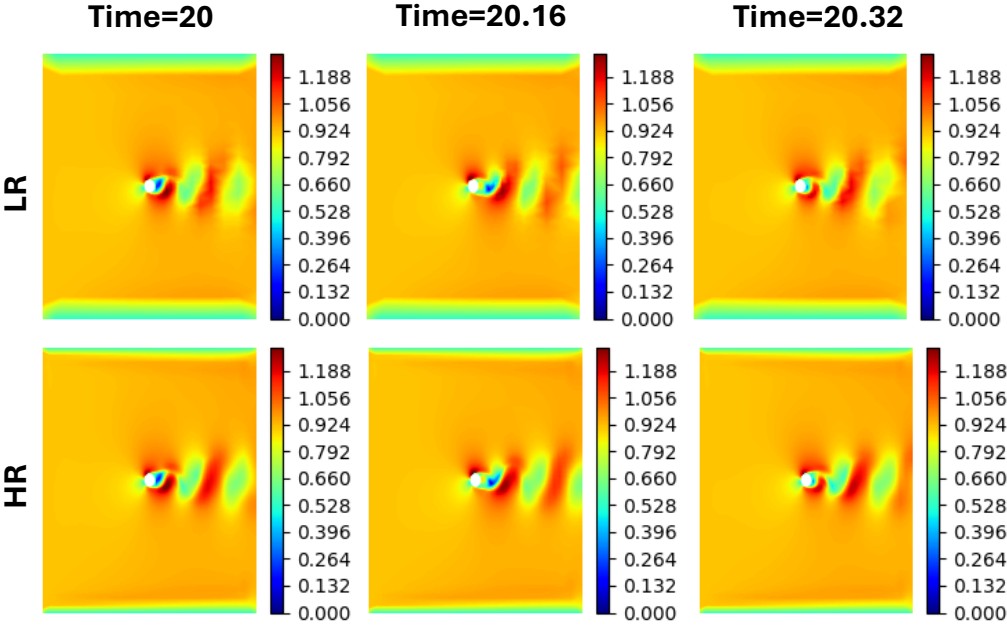

*Figure 17.* Example velocity magnitudes of LR and HR data samples corresponding to multiple timestamps from the time-dependent PDE dataset 1.

## G.6. Time-Dependent PDE Dataset 2

The time-dependent PDE dataset 2 is generated using the JAX-CFD library (Kochkov et al., 2021b), following the Kolmogorov-type forced 2D incompressible Navier–Stokes dynamics. Examples of HR and LR vorticity fields are visualized in Figure 18. In contrast to the time-dependent PDE dataset 1, this dataset is designed so that the HR and LR trajectories exhibit markedly different flow structures.

The computation domain is a periodic square domain $(0, 2\pi) \times (0, 2\pi)$. On this domain, we solve the 2D incompressible Navier–Stokes equations under external forcing:

$$\nabla \cdot \mathbf{u} = 0, \tag{25}$$

$$\frac{\partial \mathbf{u}}{\partial t} + \mathbf{u} \cdot \nabla \mathbf{u} = -\nabla p + \nu \nabla^2 \mathbf{u} + \mathbf{f}, \tag{26}$$

where $\mathbf{u}$ is velocity, $p$ is pressure, and $\nu$ is the kinematic viscosity. We set $\nu = 10^{-2}$ and maximum velocity to be 7. $\mathbf{f}$ is the external forcing term.

The HR simulation uses a $1024 \times 1024$ spectral grid, while the LR simulation uses a $32 \times 32$ grid. The HR time step is determined using the stability condition provided in JAX-CFD, and the LR time step is scaled proportionally to the grid resolution ratio. Both simulations are advanced using the Crank–Nicolson RK4 integrator implemented in the JAX-CFD spectral module.

The LR initial condition is downsampled in the Fourier space. We truncate the HR Fourier coefficients to retain only the lowest $32 \times 32$ modes and rescale the amplitudes to preserve energy. We evaluate the vorticity, $\omega = \nabla \times \mathbf{u}$, at every grid point for all time steps. A total of 300 frames are collected for both HR and LR simulations.

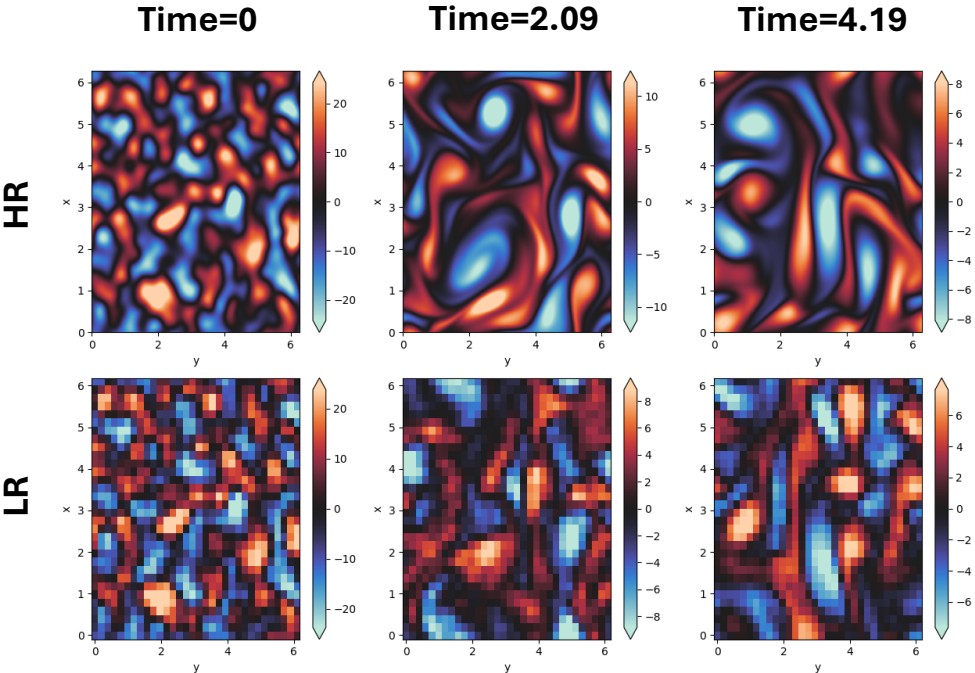

*Figure 18.* Examples of LR and HR data samples corresponding to multiple timestamps from the time-dependent PDE dataset 2.

# H. Physical Validation of Datasets

## H.1. FEM Datasets

To verify the physical validity of the three FEM datasets, we conduct mesh convergence tests on the stress and electric field in high-concentration regions. For Dataset 1, the high-concentration regions are defined as circles of radius 0.025 located to the left and right of the bottom-right hole. For Dataset 2, the region is defined as a circle of radius 0.035 located to the right of the bottom-right hole. For Dataset 3, the regions are defined as circles of radius 0.026 located to the left of the bottom-right hole and to the right of the bottom-left hole. As shown in Figure 19, these quantities exhibit clear saturation as the mesh is refined, indicating convergence toward a mesh-independent regime.

Our HR FEM datasets contain approximately 4000 nodes, which lie near the onset of this convergence plateau. While further refinement may yield slight accuracy improvements, the gain is marginal relative to the substantial increase in data generation cost. Given that our objective is to study data-efficient super-resolution, we consider this resolution regime to be appropriate. Moreover, if the HR solutions were fully mesh-independent, the super-resolution task would become trivial.

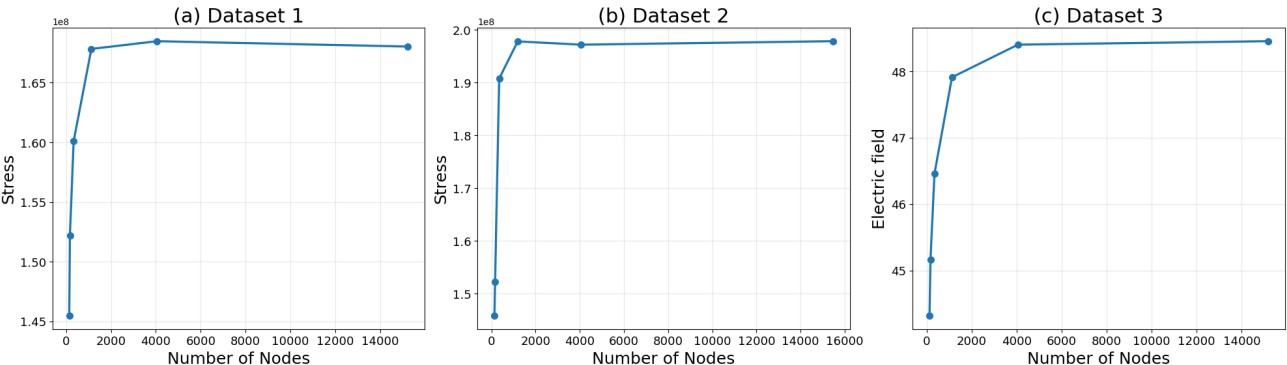

*Figure 19.* Mesh convergence tests of stress and electric field in high-concentration regions across three FEM datasets.

## H.2. Real-World Geometry Dataset

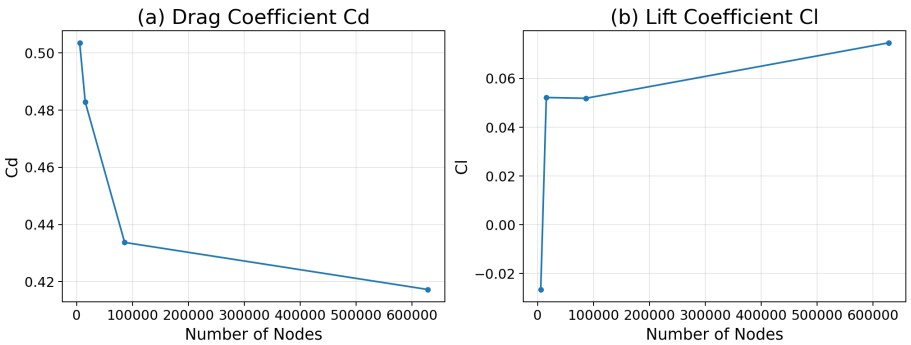

*Figure 20.* Mesh convergence tests of the drag and lift coefficients for the real-world geometry dataset.

The mesh convergence results in Figure 20 demonstrate that both drag and lift coefficients exhibit clear convergence trends with mesh refinement, verifying physical validity of the real-world geometry dataset. The HR data of the real-world geometry dataset, comprising approximately 80,000 nodes, lies near the onset of the convergence plateau.

## H.3. Time-Dependent PDE Dataset 1

To verify the physical validity of the Time-dependent PDE dataset 1, we conduct mesh convergence tests on mean and amplitude of drag coefficients and amplitude of lift coefficients, which reflect vortex shedding behind a cylinder. As depicted in Figure 21, those values approach constant values with mesh refinement. Our HR data contain approximately 20,000 nodes so that the HR data fall near the onset of the plateau regime, consistent with the FEM datasets.

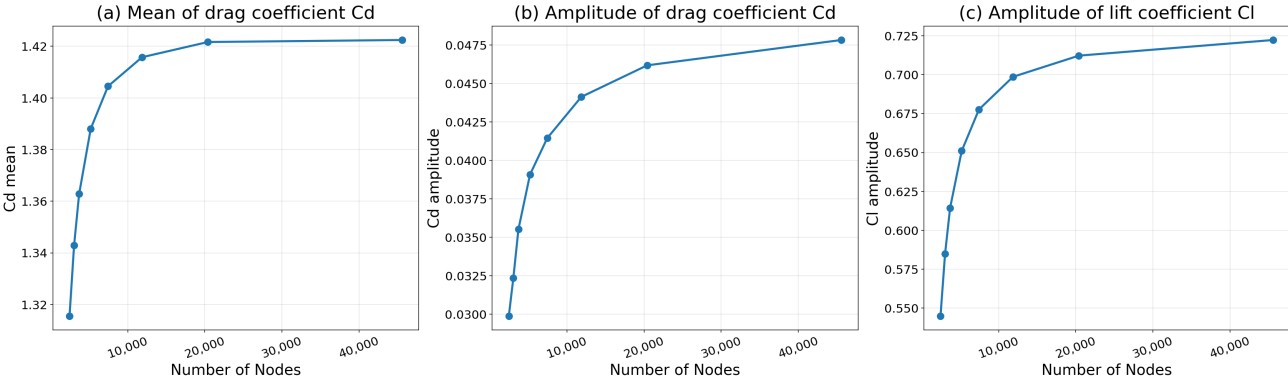

*Figure 21.* Mesh convergence tests of the mean and amplitude of drag coefficients, and the amplitude of lift coefficients, for the time-dependent PDE dataset 1.

# I. Additional Experimental Results and Analyses

The following Table 10 summarizes the contents of this section.

*Table 10.* Summary of additional experimental results and analyses.

| Contents |
| --- |
| I.1. Comparison of MAgNet, Full Supervision, and SuperMeshNet |
| I.2. Analysis on Model Architectures |
| I.3. Analysis on Inductive Biases |
| I.4. Comparison with Full Supervision |
| I.5. Computational Cost |
| I.6. Scalability |
| I.7. Comparison with Benchmark Semi-Supervised Regression Methods |
| I.8. Comparison with a Super-Resolution Competitor |
| I.9. Ablation Studies on Inductive Biases |
| I.10. Ablation Study on Core Components |
| I.11. Application to Real-World Geometry and Time-Dependent PDE |
| I.12. Analysis on HR Data Sampling Strategies |
| I.13. Training Stability |
| I.14. Analysis on $k$NN Interpolation |
| I.15. Comparison with a PINN |

### I.1. Comparison of MAgNet, Full Supervision, and **SuperMeshNet**

We empirically show the effect of HR supervision by comparing no HR supervision (MAgNet (Boussif et al., 2022)), partial supervision (SuperMeshNet), and full supervision cases. Table 11 demonstrates that MAgNet's RMSE is far apart from that of the fully supervised baseline and SuperMeshNet across all three datasets, while being approximately up to four times higher.

*Table 11.* RMSE comparison of MAgNet (Boussif et al., 2022), the baseline using fully supervised learning, and SuperMeshNet. SuperMeshNet is MGN with inductive biases trained with $N_h = 20$ and $N = 200$, and the baseline is MGN without inductive biases trained with $N_h = N = 200$. MAgNet is a zero-shot super-resolution method trained with $N_h = 0$ (*i.e.*, no HR data) and $N = 200$.

|  | Dataset 1 | Dataset 2 | Dataset 3 |
|---|---|---|---|
| MAgNet | $0.0979 \pm 0.0009$ | $0.1305 \pm 0.0007$ | $0.0754 \pm 0.0014$ |
| Fully supervised | $0.0228 \pm 0.0015$ | $0.0461 \pm 0.0004$ | $0.0243 \pm 0.0017$ |
| SuperMeshNet | $0.0226 \pm 0.0007$ | $0.0507 \pm 0.0011$ | $0.0245 \pm 0.0005$ |

## I.2. Analysis on Model Architectures

### I.2.1. EFFECT OF OUR SHARED FEATURE EXTRACTOR

As addressed in subsection 2.3, in SuperMeshNet, the two models $F_\theta$ and $G_\phi$ use a shared feature extractor. To assess its effect, we compare our default shared-extractor implementation with a variant that employs separate extractors for the two models. As shown in Table 12, using separate extractors improves accuracy, but at a substantially higher computational cost. The higher accuracy is because each extractor can specialize more strongly for the distinct roles of the two models. Therefore, the two models' pseudo-label errors are less correlated, resulting in synergy between the two. In contrast, sharing the extractor reduces training time by more than a factor of three by avoiding redundant feature extraction on the same input under complementary learning. Given this trade-off between accuracy and computational complexity, we adopt the shared feature extractor in SuperMeshNet.

*Table 12.* Comparison of a shared feature extractor with separate feature extractors in terms of RMSE and training time. Here, MGN is employed as an MPNN for each method. Training is conducted when $N_h$=20 and $N$=200 for Dataset 1.

| Methods | RMSE | Training time (s) |
|---|---|---|
| Separate feature extractors | 0.0192 | 962.85 |
| Shared feature extractor | 0.0226 | 263.72 |

### I.2.2. APPLICATION TO A CNN-BASED ARCHITECTURE OR IMAGE SUPER-RESOLUTION

To test whether our complementary learning can be generalized beyond MPNN architectures, we additionally apply complementary learning to a CNN-based architecture. Specifically, we replace both the LR and HR MPNN processors with ResNet blocks. For regular-grid data, we also replace $k$NN interpolation with bilinear interpolation for computational efficiency. Table 13 demonstrates that complementary learning not only improves HR-data efficiency but also even outperforms the fully supervised baseline on the regular-grid setting for the time-dependent PDE dataset 2. This result indicates that complementary learning is architecture-agnostic and can be flexibly extended beyond MPNNs.

*Table 13.* Application of complementary learning to a CNN-based architecture.

| Methods | $N_h$ | $N$ | RMSE |
|---|---|---|---|
| Complementary learning + ResNet | 40 | 200 | 0.0339 |
| Full supervision + ResNet | 200 | 200 | 0.0417 |

In principle, the proposed complementary learning is not inherently restricted to mesh-based simulations. At its core, the method leverages unpaired LR data samples through mutual supervision between two complementary models, which can be naturally extended to other super-resolution tasks such as image super-resolution. That being said, we would like to emphasize simulation domains, where data scarcity, corresponding to our target problem setting, is significantly more severe compared to the case of image super-resolution.

## I.3. Analysis on Inductive Biases

We observe that effects of our inductive biases depends on the task.

- For tasks where the global mean information is less important (*e.g.*, super-resolution), inductive biases tend to smooth the optimization landscape and can improve RMSE performance.

- For tasks where the global mean is important (*e.g.*, norm prediction), inductive biases removes useful information and may hinder optimization, leading to degraded performance.

In this section, we first consider super-resolution and norm prediction as representative tasks where global mean information is, respectively, less important and more important. We further present additional exemplary tasks that may benefit from our inductive biases. Furthermore, we compare our inductive biases with standard normalization techniques.

### I.3.1. EXAMPLE 1: SUPER-RESOLUTION

**Task description**

The goal of super-resolution is to predict HR fields from LR fields.

**Importance of the global mean on the super-resolution task**

As summarized in Table 14, subtracting the mean from LR data (i.e., using the deviation from the mean as input) results in only a minor change in RMSE, whereas providing only the global mean (i.e., using the mean of the LR data as input) significantly degrades performance. This suggests that the global mean contributes little to prediction accuracy. Super-resolution primarily involves learning high-frequency discrepancies between LR and HR fields (Guo et al., 2025). Moreover, our architecture (Figures 11 and 12) generates HR predictions by refining a $k$NN-interpolated coarse estimate, which may further reduce the need to model the global mean explicitly.

*Table 14.* RMSE comparison of MGN-based SuperMeshNet trained for super-resolution task on Dataset 1, using the following three cases as input: the LR data sample, its deviation from the mean, and the mean.

| Input | RMSE |
|---|---|
| LR data | 0.0228 |
| Deviation of LR data | 0.0235 |
| Mean of LR data | 0.0617 |

We further clarify that whether mean information matters for a task is determined by how strongly the ground truth output is influenced by the mean of the input. In Figure 22, the super-resolution task does not exhibits a clear dependency between the input mean and the target output, demonstrating that global mean information is relatively unimportant for this task.

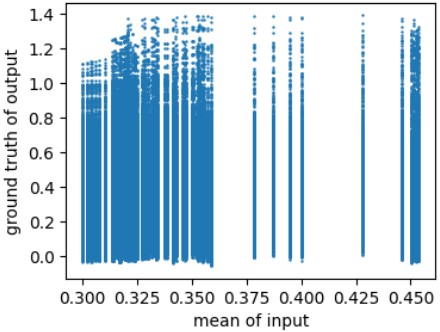

*Figure 22.* Relationship between the mean of the input and the ground truth output for the super-resolution task.

**Effect of inductive biases on loss landscape**

As illustrated in Figure 23, compared to the case without our inductive biases (w/o IB), the cases with our inductive biases (w IB) yield a smoother loss landscape over training iterations in the super-resolution task. As reported in prior work on batch normalization (Santurkar et al., 2018), a smoother loss landscape stabilizes and improves optimization, possibly improving the performance.

To measure the loss landscape, following the prior work in (Santurkar et al., 2018), we perturb the model parameters by taking a step along the gradient direction and evaluate the loss at the perturbed point. The perturbation magnitude is set to a fixed multiple of the learning rate (four times the learning rate in our experiments).

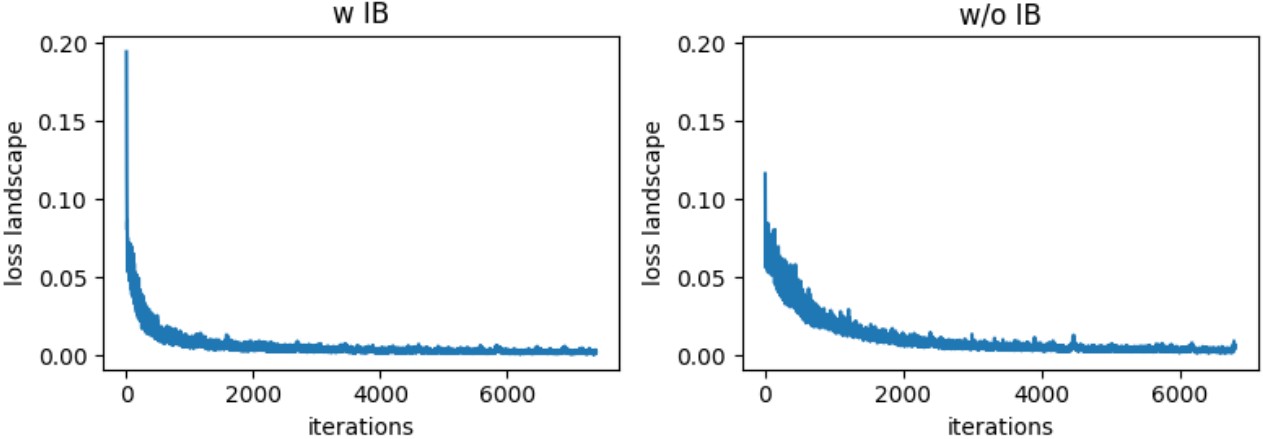

*Figure 23.* Effect of inductive biases on the loss landscape for the super-resolution task when SuperMeshNet with and without inductive biases (corresponding to w IB and w/o IB, respectively) is used. Here, MGN is employed as an MPNN for each method and is trained when $N_h$=20 and $N$=200 for Dataset 1.

**Effect of inductive biases on RMSE performance**

As summarized in Table 15, our inductive bias empirically improves RMSE performance in super-resolution. Although it removes global mean information, this information is relatively less important for super-resolution tasks. Moreover, the inductive bias smooths the loss landscape, facilitating improved performance.

*Table 15.* Effect of inductive biases on the performance of super-resolution task. All experiments are conducted using MGN-based SuperMeshNet and Dataset 1.

| Inductive biases | RMSE |
|:---:|:---:|
| O | 0.0226 |
| X | 0.0269 |

I.3.2. EXAMPLE 2: NORM PREDICTION

**Task description**

We consider an exemplary task, whose goal is to predict the (normalized) norm of LR input fields, which is constant across nodes.

**Importance of the global mean on the norm prediction task**

For the norm prediction, the global mean appears to contain essential information. As summarized in Table 16, when only the mean is provided as input, performance gets improved, whereas providing only the deviation significantly degrades performance.

*Table 16.* RMSE comparison of MGN-based SuperMeshNet trained for norm prediction task on Dataset 1, using the following three cases as input: the LR data sample, its deviation from the mean, and the mean.

| Input | RMSE |
| --- | --- |
| LR data | 0.00269 |
| Deviation of LR data | 0.00378 |
| Mean of LR data | 0.00149 |

Furthermore, as illustrated in Figure 24, the norm prediction task exhibits a clear dependency between the input mean and the target output (the norm), indicating that the mean indeed carries essential information.

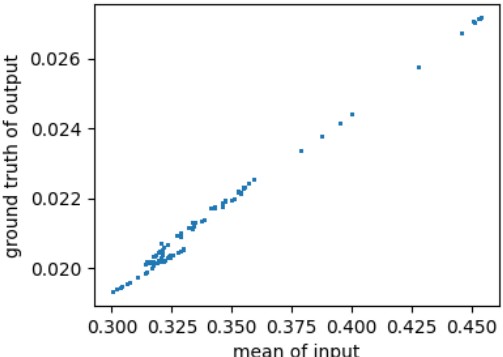

*Figure 24.* Relationship between the mean of the input and the ground truth output for the norm prediction task.

**Effect of inductive biases on loss landscape**

As illustrated in Figure 25, the case with inductive biases (w IB) yields a noticeably rougher loss landscape in norm prediction.

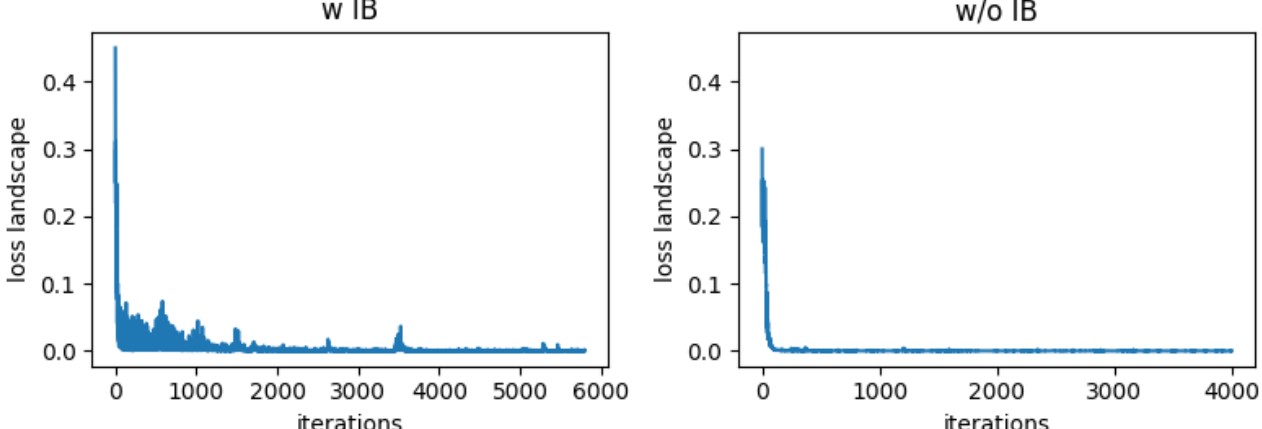

*Figure 25.* Effect of inductive biases on the loss landscape for the norm prediction task when SuperMeshNet with and without inductive biases (corresponding to w IB and w/o IB, respectively) is used. Here, MGN is employed as an MPNN for each method and is trained when $N_h$=20 and $N$=200 for Dataset 1.

**Effect of inductive biases on RMSE performance**

As summarized in Table 17, our inductive bias significantly degrades accuracy in norm prediction. This is because it removes essential information—the global mean—and makes the loss landscape rougher, resulting in challenging optimization.

*Table 17.* Effect of inductive biases on the performance of norm prediction task. All experiments are conducted using MGN-based SuperMeshNet and Dataset 1.

| Inductive biases | RMSE |
|---|---|
| O | 0.00924 |
| X | 0.00647 |

I.3.3. FURTHER EXAMPLES

Below, we additionally provide concrete examples of tasks where the mean is typically less important:

1) Scientific tasks

- Vorticity prediction: the vorticity field depends on the local rotational structure, not the mean velocity.

- Force prediction: forces depend on relative distances, not absolute positions.

2) General graph learning tasks

- Node classification: predictions rely mainly on relational structures, not absolute feature scales.

- Link prediction: predictions depend on similarity between features rather than the absolute feature scale.

These tasks may benefit from our inductive biases.

### I.3.4. COMPARISON WITH STANDARD NORMALIZATION

*Table 18.* Comparison of our inductive biases in SuperMeshNet with layer normalization (Ba et al., 2016) and batch normalization (Ioffe & Szegedy, 2015). All models are based on the MGN architecture and trained using complementary learning with $N_h = 20$ and $N = 200$. The best-performing result in each case is highlighted in **bold**.

| Dataset | Methods | RMSE |
|:---:|:---:|:---:|
| 1 | Layer normalization (Ba et al., 2016) | $0.0310 \pm 0.0014$ |
| | Batch normalization (Ioffe & Szegedy, 2015) | $\textbf{0.0165} \pm 0.0009$ |
| | SuperMeshNet | $0.0226 \pm 0.0007$ |
| 2 | Layer normalization (Ba et al., 2016) | $0.0587 \pm 0.0008$ |
| | Batch normalization (Ioffe & Szegedy, 2015) | $0.0508 \pm 0.0007$ |
| | SuperMeshNet | $\textbf{0.0507} \pm 0.0011$ |
| 3 | Layer normalization (Ba et al., 2016) | $0.0297 \pm 0.0013$ |
| | Batch normalization (Ioffe & Szegedy, 2015) | $0.0250 \pm 0.0003$ |
| | SuperMeshNet | $\textbf{0.0245} \pm 0.0005$ |

Similarly as in our inductive biases, existing normalization techniques such as layer normalization (Ba et al., 2016) and batch normalization (Ioffe & Szegedy, 2015) also involve mean subtraction. In the context of MPNNs, layer normalization computes the mean across features within each node embedding, whereas batch normalization computes the mean across node embeddings, which is also leveraged in our inductive biases. Thus, rather than layer normalization (Ba et al., 2016), our inductive biases are closer to batch normalization (Ioffe & Szegedy, 2015). Standard normalization techniques typically include additional operations such as scaling by the standard deviation and the use of learnable shift and scale parameters. The results in Table 18 demonstrate that, without these additional components, SuperMeshNet consistently outperforms layer normalization and, in some cases, even surpasses batch normalization. This suggests that centering alone, namely mean subtraction, acts as a crucial inductive bias for the super-resolution task.

Although our inductive biases are rather simple and closely related to conventional normalization methods, they offer several key insights. First, computing the mean *across* nodes (as in batch normalization and SuperMeshNet) is proven to be more effective than computing the mean *within* each node (as in layer normalization), likely because super-resolution benefits more from local deviations from the global mean. Second, the additional components of standard normalization, such as division by the standard deviation and learnable affine parameters, do not necessarily yield performance gains. This underscores that centering itself is the most essential component for super-resolution tasks.

## I.4. Comparison with Full Supervision

### I.4.1. RMSE

In Table 19, we present the full version of Table 4 from the main manuscript, now including the standard deviation of RMSE values. In some cases, SuperMeshNet ($N_h$=20, $N$=200) yields slightly worse RMSEs than the fully supervised baselines ($N_h$=$N$=200). In such cases, Table 20 summarizes additional results using SuperMeshNet with $N_h$=40 and $N = 200$, where $N_h$ is set to a slightly larger value than our default setting (*i.e.*, $N_h = 20$) but still significantly smaller than 200, in comparison with the fully supervised ($N_h$=$N$=200) baselines. These results demonstrate that using only 20% of HR data samples (*i.e.*, $N_h = 40$) in SuperMeshNet is sufficient to outperform the fully supervised baseline.

*Table 19.* The RMSE of SuperMeshNet (with inductive biases) and SuperMeshNet-O (without inductive biases) trained with $N_h = 20$ HR data samples and $N = 200$ LR data samples across six MPNNs and three datasets, in comparison with two fully supervised MPNNs including 1) $N_h = N = 20$ and 2) $N_h = N = 200$. The best performer is highlighted in **bold**.

| | Method | MPNN | | | | | |
|---|---|---|---|---|---|---|---|
| | $(N_h, N)$ | GCN | SAGE | GAT | GTR | GIN | MGN |
| Dataset 1 | Fully supervised (20, 20) | 0.0874 ± 0.0039 | 0.0876 ± 0.0015 | 0.0826 ± 0.0042 | 0.0758 ± 0.0068 | 0.0819 ± 0.0047 | 0.0655 ± 0.0030 |
| | Fully supervised (200, 200) | 0.0575 ± 0.0035 | 0.0544 ± 0.0025 | 0.0512 ± 0.0016 | 0.0450 ± 0.0023 | 0.0381 ± 0.0027 | 0.0228 ± 0.0015 |
| | SuperMeshNet-O (20, 200) | 0.0613 ± 0.0020 | 0.0589 ± 0.0021 | 0.0544 ± 0.0008 | 0.0451 ± 0.0020 | 0.0404 ± 0.0028 | 0.0269 ± 0.0019 |
| | SuperMeshNet (20, 200) | **0.0431** ± 0.0009 | **0.0450** ± 0.0010 | **0.0457** ± 0.0016 | **0.0385** ± 0.0029 | **0.0277** ± 0.0006 | **0.0226** ± 0.0007 |
| Dataset 2 | Fully supervised (20, 20) | 0.0972 ± 0.0082 | 0.1025 ± 0.0052 | 0.0983 ± 0.0026 | 0.0983 ± 0.0016 | 0.0775 ± 0.0073 | 0.0730 ± 0.0075 |
| | Fully supervised (200, 200) | 0.0624 ± 0.0022 | 0.0633 ± 0.0032 | 0.0637 ± 0.0013 | **0.0572** ± 0.0016 | **0.0534** ±0.0009 | **0.0461** ± 0.0004 |
| | SuperMeshNet-O (20, 200) | 0.0636 ± 0.0013 | 0.0664 ± 0.0032 | 0.0680 ± 0.0023 | 0.0631 ± 0.0018 | 0.0569 ± 0.0023 | 0.0514 ± 0.0003 |
| | SuperMeshNet (20, 200) | **0.0574** ± 0.0003 | **0.0624** ± 0.0006 | **0.0634** ± 0.0018 | 0.0600 ± 0.0012 | 0.0537 ± 0.0015 | 0.0507 ± 0.0011 |
| Dataset 3 | Fully supervised (20, 20) | 0.0587 ± 0.0038 | 0.0611 ± 0.0043 | 0.0616 ± 0.0050 | 0.0513 ± 0.0052 | 0.0569 ± 0.0016 | 0.0523 ± 0.0055 |
| | Fully supervised (200, 200) | 0.0370 ± 0.0029 | 0.0340 ± 0.0015 | 0.0374 ± 0.0012 | 0.0329 ± 0.0021 | 0.0317 ± 0.0022 | **0.0243** ± 0.0017 |
| | SuperMeshNet-O (20, 200) | 0.0380 ± 0.0018 | 0.0366 ± 0.0021 | 0.0375 ± 0.0009 | 0.0363 ± 0.0023 | 0.0316 ± 0.0010 | 0.0281 ± 0.0006 |
| | SuperMeshNet (20, 200) | **0.0297** ± 0.0008 | **0.0297** ± 0.0014 | **0.0310** ± 0.0012 | **0.0294** ± 0.0011 | **0.0258** ± 0.0008 | 0.0245 ± 0.0005 |

*Table 20.* The RMSE and its standard deviation of SuperMeshNet trained with $N_h = 40$ HR data samples and $N = 200$ LR data samples, in comparison with a fully supervised MPNN trained with $N_h = N = 200$. Experiments are conducted only for cases where using $N_h = 20$ is insufficient to outperform the fully supervised baseline.

| Dataset | Method | $N_h$ | $N$ | MPNN | RMSE |
|---|---|---|---|---|---|
| 2 | SuperMeshNet | 40 | 200 | GTR | 0.0568 ± 0.0015 |
| 2 | Fully supervised | 200 | 200 | GTR | 0.0572 ± 0.0016 |
| 2 | SuperMeshNet | 40 | 200 | GIN | 0.0501 ± 0.0004 |
| 2 | Fully supervised | 200 | 200 | GIN | 0.0534 ± 0.0009 |
| 2 | SuperMeshNet | 40 | 200 | MGN | 0.0461 ± 0.0011 |
| 2 | Fully supervised | 200 | 200 | MGN | 0.0461 ± 0.0004 |
| 3 | SuperMeshNet | 40 | 200 | MGN | 0.0225 ± 0.0003 |
| 3 | Fully supervised | 200 | 200 | MGN | 0.0243 ± 0.0017 |

### I.4.2. PHYSICS-AWARE EVALUATION

We compare SuperMeshNet with a fully supervised baseline by evaluating prediction within regions of interest, rather than averaging over the entire domain. These regions correspond to areas where stress and electric field are highly concentrated. For Dataset 1, the high-concentration regions are defined as circles of radius 0.025 located to the left and right of the bottom-right hole. For Dataset 2, the region is defined as a circle of radius 0.035 located to the right of the bottom-right hole. For Dataset 3, the regions are defined as circles of radius 0.026 located to the left of the bottom-right hole and to the right of the bottom-left hole. As shown in Table 21, SuperMeshNet predicts stress and electric field in these concentrated regions with less than 5% relative error, while outperforming the fully supervised baseline despite using fewer HR samples.

*Table 21.* Comparison between SuperMeshNet, trained with $N_h = 40$ high-resolution (HR) samples and $N = 200$ low-resolution (LR) samples, and a fully supervised MPNN trained with $N_h = N = 200$, in terms of stress prediction (Datasets 1 and 2) and electric field prediction in highly concentrated regions.

| Methods | Dataset 1 Stress (relative error) | Dataset 2 Stress (relative error) | Dataset 3 Electric field (relative error) |
|---|---|---|---|
| Ground truth HR data | $1.65 \times 10^8$ | $2.00 \times 10^8$ | 47.4 |
| SuperMeshNet ($N_h = 40, N = 200$) | $1.70 \times 10^8$ (0.0255) | $1.98 \times 10^8$ (0.0100) | 45.4 (0.0429) |
| Fully supervised ($N = N_h = 200$) | $1.72 \times 10^8$ (0.0382) | $1.95 \times 10^8$ (0.0223) | 43.8 (0.0770) |

## I.5. Computational Cost

### I.5.1. TRAINING COST

SuperMeshNet alleviates the reliance on expensive HR data samples but incurs longer training time compared to the fully supervised baseline. Figures 26–28 display how the training time increases and data generation time decreases—resulting from the use of SuperMeshNet ($N_h = 20$, $N = 200$)—scale with mesh size, relative to fully supervised learning ($N_h = N = 200$). To estimate the data generation time decrease, we measure the time required to generate 180 HR data samples by solving the PDE using a direct solver on an Intel(R) Core(TM) i7-9700K CPU @ 3.60GHz. For the training time increase, we compute the difference between the training times of MGN under fully supervised learning and SuperMeshNet, using an Intel (R) Core (TM) i9-10920X CPUs@3.50 GHz and an NVIDIA RTX A6000 GPU. The slopes in each figure are computed using the least square method. Comparison of the two slopes, which characterize how training time increases and data generation time decreases with mesh size, respectively, consistently reveals that, on all three datasets, data generation time grows more rapidly as mesh size decreases.

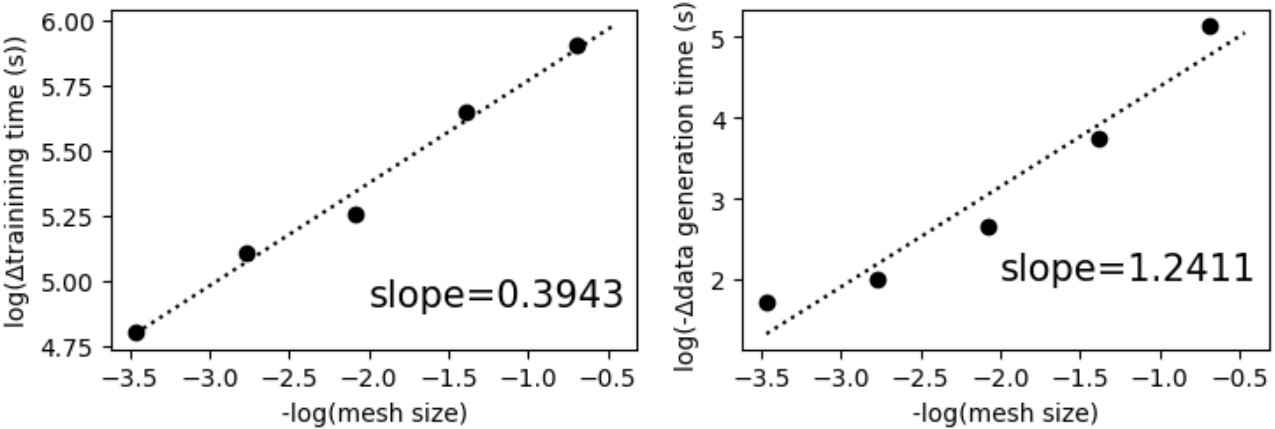

*Figure 26.* Training time increase (left) and data generation time decrease (right), resulting from the use of SuperMeshNet ($N_h = 20$, $N = 200$), relative to fully supervised learning ($N_h = N = 200$) on Dataset 1 and its mesh-size variants. All experiments use MGN as the underlying MPNN architecture.

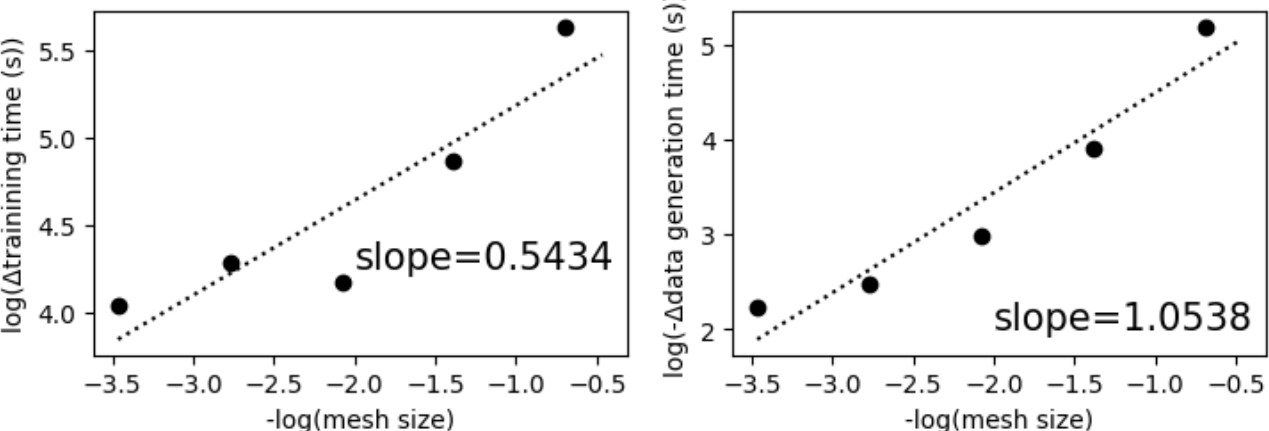

*Figure 27.* Training time increase (left) and data generation time decrease (right), resulting from the use of SuperMeshNet ($N_h = 20$, $N = 200$), relative to fully supervised learning ($N_h = N = 200$) on Dataset 2 and its mesh-size variants. All experiments use MGN as the underlying MPNN architecture.

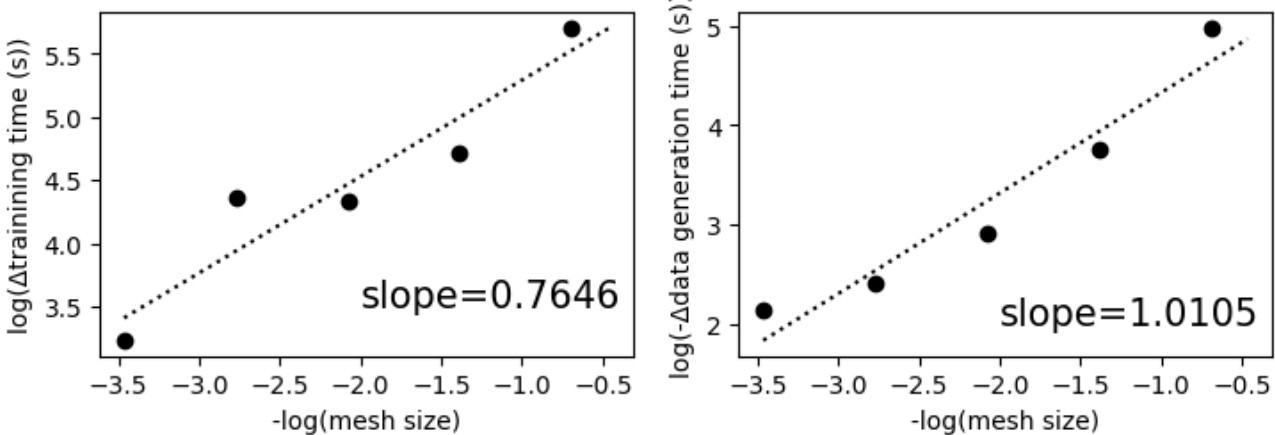

*Figure 28.* Training time increase (left) and data generation time decrease (right), resulting from the use of SuperMeshNet ($N_h = 20$, $N = 200$), relative to fully supervised learning ($N_h = N = 200$) on Dataset 3 and its mesh-size variants. All experiments use MGN as the underlying MPNN architecture.

### I.5.2. INFERENCE COST

Figure 29 demonstrates that SuperMeshNet substantially reduces the computational cost of HR simulation. Specifically, the combined computational cost of LR simulation and subsequent SuperMeshNet inference is significantly lower than the cost of HR simulation. The time saving becomes even more pronounced for small mesh sizes, where the computational cost required for HR simulation grows rapidly.

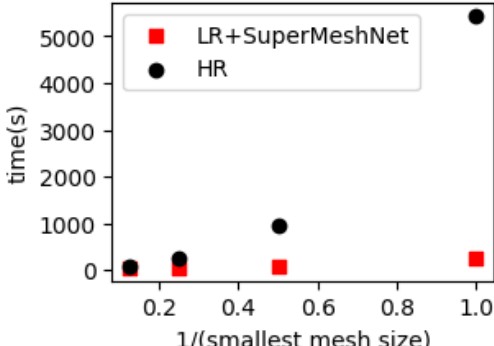

*Figure 29.* Comparison of the HR simulation time with the combined computational cost of LR simulation and subsequent SuperMesh-Net's inference. Each value is measured over 1,000 data samples.

### I.6. Scalability

#### I.6.1. MESH SIZE AND MAGNIFICATION RATIO

In Table 22, we evaluate the scalability of our framework SuperMeshNet by measuring the training time and RMSE as a function of mesh size (while fixing the magnification ratio). These experiments are conducted using MGN, trained with 40 HR and 200 LR data samples from Dataset 1. According to the results, the training time increases moderately as the mesh becomes finer. The RMSE tends to increase and then decrease again as the mesh becomes extremely fine. This is because we fix the magnification ratio (the ratio between the LR mesh size and the HR mesh size) while varying both the LR mesh size and the HR mesh size. When the mesh becomes finer, the HR data samples contain more detailed features, making the LR-to-HR transformation more challenging. However, once the mesh resolution exceeds the characteristic length scale of the domain geometry, no additional details can be represented in the HR mesh. At the same time, the LR data samples already capture most of the meaningful features, and the LR-to-HR transformation becomes easier again. For comparison, we provide Table 23, presenting experimental results where the LR mesh size is fixed while the magnification ratio (the HR mesh size) varies. In this case, the RMSE tends to monotonically increase as the HR mesh becomes finer since the LR mesh size is fixed.

*Table 22.* Training time (in seconds) and RMSE of MGN-based SuperMeshNet trained with 40 HR and 200 LR data samples from Dataset 1 as a function of mesh size while fixing the magnification ratio.

| Smallest mesh size | Training time (s) | RMSE |
|---|---|---|
| 0.016 | 383.28 | 0.0112 |
| 0.008 | 443.34 | 0.0158 |
| 0.004 | 587.50 | 0.0194 |
| 0.002 | 710.96 | 0.0121 |

*Table 23.* RMSE of MGN-based SuperMeshNet trained with 40 HR and 200 LR data samples from Dataset 1 as a function of magnification ratio (HR mesh size) while fixing the LR mesh size.

| Magnification ratio | RMSE |
|---|---|
| $2 \times 2$ | 0.0099 |
| $4 \times 4$ | 0.0112 |
| $8 \times 8$ | 0.0689 |
| $16 \times 16$ | 0.0796 |

#### I.6.2. LARGE DATASET

We further validate scalability on BlastNet 2.0 (Chung et al., 2023), which contains approximately 2 million nodes. We conduct experiments on two subsets (Forced Hit and Parametric Variation) of BlastNet 2.0, which include regular grid-based multi-physics (reaction–flow) and turbulent flow simulations. For the experiments, we set the hidden dimension to 10 and use two MPNN layers for both the LR processor and the HR processor. The results in Table 24 verify the scalability of SuperMeshNet on large-scale datasets with over one million nodes.

*Table 24.* RMSE comparison between a fully supervised baseline and SuperMeshNet on two subsets of BlastNet 2.0 (Chung et al., 2023).

| Sub-dataset | Fully supervised ($N_h{=}N{=}50$) | SuperMeshNet ($N_h{=}10$ $N{=}50$) |
|---|---|---|
| Forced hit | 0.0451 | 0.0230 |
| Parametric variation | 0.0785 | 0.0643 |

## I.7. Comparison with Benchmark Semi-Supervised Regression Methods

### I.7.1. ACCURACY AND TRAINING TIME

Table 25 presents the full version of Table 5 from the main manuscript, now including the mean and standard deviation of RMSE and training time for all datasets under the setting of $N_h = 20$ HR data samples and $N = 200$ LR data samples. SuperMeshNet consistently results in the shortest training time across all datasets, in comparison with benchmark semi-supervised regression methods. It also achieves the lowest RMSE on Datasets 1 and 3. For Dataset 2, while SuperMeshNet yields a slightly higher RMSE, it significantly reduces training time compared to all the benchmarks. Overall, SuperMeshNet is shown to reveal strong potential in terms of both predictive accuracy and training efficiency. In Table 26, we additionally report that, for Dataset 2 with $N_h = 40$ and $N = 200$, SuperMeshNet still exhibits both the lowest RMSE and the shortest training time among all evaluated methods.

*Table 25.* Comparison with benchmark semi-supervised regression methods in terms of the RMSE and training time (in seconds). Here, MGN is employed as an MPNN architecture for each method. Training is conducted when $N_h = 20$ and $N = 200$ for each dataset. The best value in each metric is highlighted in **bold**.

| Dataset | Methods | RMSE | Training time (s) |
|---|---|---|---|
| 1 | Mean-Teacher (Tarvainen & Valpola, 2017) | $0.0325 \pm 0.0016$ | $694 \pm 50$ |
| | TNNR (Wetzel et al., 2022) | $0.0624 \pm 0.0202$ | $477 \pm 333$ |
| | UCVME (Dai et al., 2023) | $0.0293 \pm 0.0012$ | $1123 \pm 102$ |
| | SuperMeshNet-O | $0.0269 \pm 0.0019$ | $503 \pm 64$ |
| | SuperMeshNet | $\mathbf{0.0226} \pm 0.0007$ | $\mathbf{421} \pm 93$ |
| 2 | Mean-Teacher (Tarvainen & Valpola, 2017) | $0.0499 \pm 0.0007$ | $402 \pm 28$ |
| | TNNR (Wetzel et al., 2022) | $0.0823 \pm 0.0055$ | $350 \pm 90$ |
| | UCVME (Dai et al., 2023) | $\mathbf{0.0484} \pm 0.0006$ | $739 \pm 59$ |
| | SuperMeshNet-O | $0.0514 \pm 0.0003$ | $306 \pm 16$ |
| | SuperMeshNet | $0.0507 \pm 0.0011$ | $\mathbf{250} \pm 9$ |
| 3 | Mean-Teacher (Tarvainen & Valpola, 2017) | $0.0270 \pm 0.0003$ | $446 \pm 33$ |
| | TNNR (Wetzel et al., 2022) | $0.0393 \pm 0.0027$ | $437 \pm 83$ |
| | UCVME (Dai et al., 2023) | $0.0281 \pm 0.0013$ | $700 \pm 89$ |
| | SuperMeshNet-O | $0.0281 \pm 0.0006$ | $345 \pm 10$ |
| | SuperMeshNet | $\mathbf{0.0245} \pm 0.0005$ | $\mathbf{286} \pm 18$ |

*Table 26.* Comparison with benchmark semi-supervised regression methods in terms of the RMSE and training time (in seconds). Here, MGN is employed as an MPNN architecture for each method. Training is conducted when $N_h = 40$ and $N = 200$ for each dataset. The best value in each metric is highlighted in **bold**.

| Dataset | Methods | RMSE | Training time (s) |
|---|---|---|---|
| 2 | Mean-Teacher (Tarvainen & Valpola, 2017) | $0.0474 \pm 0.0007$ | $427 \pm 37$ |
| | TNNR (Wetzel et al., 2022) | $0.0807 \pm 0.0074$ | $409 \pm 143$ |
| | UCVME (Dai et al., 2023) | $0.0474 \pm 0.0013$ | $780 \pm 89$ |
| | SuperMeshNet-O | $0.0479 \pm 0.0005$ | $348 \pm 15$ |
| | SuperMeshNet | $\mathbf{0.0461} \pm 0.0011$ | $\mathbf{292} \pm 25$ |

### I.7.2. PREDICTION DIVERSITY

The results in Table 27 demonstrate that SuperMeshNet exhibits consistently greater prediction diversity across all datasets, compared to the case of UCVME (Dai et al., 2023). The prediction diversity is quantified as the root mean square of the difference between two predictions made by MGN model pairs, each trained with $N_h = 20$ and $N = 200$ on each dataset. This implies that our architectural design for complementary learning apparently promotes greater diversity in the learning process than the case of UCVME.

*Table 27.* Comparison of prediction diversity between UCVME and SuperMeshNet. The prediction diversity is quantified as the root mean square of the difference between two predictions made by MGN model pairs trained by each method with $N_h$=20 and $N$=200 on each dataset. The dropout probability in UCVME is set to 0.1.

| Dataset | Methods | Prediction diversity |
|---------|---------|----------------------|
| 1 | UCVME (Dai et al., 2023) | $0.0017 \pm 0.0001$ |
|   | SuperMeshNet | $0.0200 \pm 0.0014$ |
| 2 | UCVME (Dai et al., 2023) | $0.0024 \pm 0.0002$ |
|   | SuperMeshNet | $0.0514 \pm 0.0011$ |
| 3 | UCVME (Dai et al., 2023) | $0.0014 \pm 0.0001$ |
|   | SuperMeshNet | $0.0294 \pm 0.0004$ |

### I.8. Comparison with a Super-Resolution Competitor

Although the primary objective of SuperMeshNet is to improve super-resolution performance across a wide range of MPNNs rather than to outperform a specific state-of-the-art method, we compare a special case of SuperMeshNet using MGN with the most recent and relevant benchmarks, SRGNN (Barwey et al., 2024), to further validate its effectiveness. The results in Table 28 signify that SuperMeshNet, even when trained with only 20 HR data samples, outperforms SRGNN (Barwey et al., 2024) trained with 200 HR data samples, underscoring its superior data efficiency.

*Table 28.* The RMSE and its standard deviation of MGN-based SuperMeshNet trained with varying numbers of HR data samples $N_h$ and a fixed $N=200$ LR data samples in comparison with SRGNN with full supervision ($N=N_h=200$).

| Dataset | Methods | $N_h$ | RMSE |
|---|---|---|---|
| 1 | SuperMeshNet | 5 | $0.0447 \pm 0.0010$ |
| | | 10 | $0.0280 \pm 0.0013$ |
| | | 20 | $0.0226 \pm 0.0007$ |
| | | 40 | $0.0191 \pm 0.0021$ |
| | SRGNN (Barwey et al., 2024) | 200 | $0.0247 \pm 0.0013$ |
| 2 | SuperMeshNet | 5 | $0.0723 \pm 0.0018$ |
| | | 10 | $0.0645 \pm 0.0022$ |
| | | 20 | $0.0507 \pm 0.0011$ |
| | | 40 | $0.0461 \pm 0.0011$ |
| | SRGNN (Barwey et al., 2024) | 200 | $0.0487 \pm 0.0011$ |
| 3 | SuperMeshNet | 5 | $0.0353 \pm 0.0015$ |
| | | 10 | $0.0294 \pm 0.0010$ |
| | | 20 | $0.0245 \pm 0.0005$ |
| | | 40 | $0.0225 \pm 0.0003$ |
| | SRGNN (Barwey et al., 2024) | 200 | $0.0254 \pm 0.0011$ |

## I.9. Ablation Studies on Inductive Biases

Table 29 presents the full version of Table 6 from the main manuscript, now including the mean and standard deviation of RMSE values for all datasets. On Dataset 2, the combination of both inductive biases (N+M) occasionally results in a higher RMSE than using a single inductive bias (N). To further investigate this, Table 30 reports additional results on Dataset 2 using SuperMeshNet with $N_h = 80$ and $N = 200$, where $N_h$ is set to a slightly larger value than our default setting (*i.e.*, $N_h = 20$) yet still significantly smaller than 200. With the larger $N_h$, the incorporation of both inductive biases consistently yields the lowest RMSE among the four inductive bias settings. These findings reveal the existence of a dataset-dependent threshold for $N_h$ above which leveraging both inductive biases becomes beneficial.

*Table 29.* Ablation studies on inductive biases. The RMSE of SuperMeshNet across six MPNNs under four inductive bias conditions (O: without inductive biases, N: node-level centering, M: message-level centering, and N+M: both node-level and message-level centering operations) trained with $N_h = 20$ and $N = 200$ for each dataset is compared. For each MPNN, the lowest RMSE value among the four inductive bias conditions is highlighted in **bold**.

| Dataset | MPNN | RMSE | | | |
| --- | --- | --- | --- | --- | --- |
| | | O | N | M | N + M |
| 1 | GCN | $0.0613 \pm 0.0020$ | $\mathbf{0.0431} \pm 0.0009$ | - | - |
| | SAGE | $0.0589 \pm 0.0021$ | $0.0493 \pm 0.0024$ | $0.0528 \pm 0.0018$ | $\mathbf{0.0450} \pm 0.0010$ |
| | GAT | $0.0544 \pm 0.0008$ | $\mathbf{0.0457} \pm 0.0016$ | - | - |
| | GTR | $0.0451 \pm 0.0020$ | $0.0405 \pm 0.0025$ | $0.0438 \pm 0.0010$ | $\mathbf{0.0385} \pm 0.0029$ |
| | GIN | $0.0404 \pm 0.0028$ | $0.0290 \pm 0.0026$ | $0.0281 \pm 0.0015$ | $\mathbf{0.0277} \pm 0.0006$ |
| | MGN | $0.0269 \pm 0.0019$ | $0.0237 \pm 0.0010$ | $0.0247 \pm 0.0014$ | $\mathbf{0.0226} \pm 0.0007$ |
| 2 | GCN | $0.0636 \pm 0.0013$ | $\mathbf{0.0574} \pm 0.0003$ | - | - |
| | SAGE | $0.0664 \pm 0.0032$ | $\mathbf{0.0623} \pm 0.0005$ | $0.0652 \pm 0.0009$ | $0.0624 \pm 0.0006$ |
| | GAT | $0.0680 \pm 0.0023$ | $\mathbf{0.0634} \pm 0.0018$ | - | - |
| | GTR | $0.0631 \pm 0.0018$ | $0.0607 \pm 0.0009$ | $0.0617 \pm 0.0025$ | $\mathbf{0.0600} \pm 0.0012$ |
| | GIN | $0.0569 \pm 0.0023$ | $\mathbf{0.0523} \pm 0.0009$ | $0.0549 \pm 0.0008$ | $0.0537 \pm 0.0015$ |
| | MGN | $0.0514 \pm 0.0003$ | $\mathbf{0.0488} \pm 0.0013$ | $0.0509 \pm 0.0004$ | $0.0507 \pm 0.0011$ |
| 3 | GCN | $0.0380 \pm 0.0018$ | $\mathbf{0.0297} \pm 0.0008$ | - | - |
| | SAGE | $0.0366 \pm 0.0021$ | $0.0309 \pm 0.0018$ | $0.0346 \pm 0.0018$ | $\mathbf{0.0297} \pm 0.0014$ |
| | GAT | $0.0375 \pm 0.0009$ | $\mathbf{0.0310} \pm 0.0012$ | - | - |
| | GTR | $0.0363 \pm 0.0023$ | $0.0312 \pm 0.0008$ | $0.0327 \pm 0.0006$ | $\mathbf{0.0294} \pm 0.0011$ |
| | GIN | $0.0316 \pm 0.0010$ | $0.0261 \pm 0.0002$ | $0.0268 \pm 0.0007$ | $\mathbf{0.0258} \pm 0.0008$ |
| | MGN | $0.0281 \pm 0.0006$ | $\mathbf{0.0245} \pm 0.0003$ | $0.0246 \pm 0.0006$ | $\mathbf{0.0245} \pm 0.0005$ |

*Table 30.* Additional ablation studies on inductive biases. The RMSE of SuperMeshNet across six MPNNs under four inductive bias conditions (O: without inductive biases, N: node-level centering, M : message-level centering, and N+M: both node-level and message-level centering operations) trained with $N_h = 80$ and $N = 200$ for Dataset 2 is compared. For each MPNN, the lowest RMSE value among the four inductive bias conditions is highlighted in **bold**.

| Dataset | MPNN | RMSE | | | |
| --- | --- | --- | --- | --- | --- |
| | | O | N | M | N + M |
| 2 | GCN | $0.0630 \pm 0.0017$ | $\mathbf{0.0556} \pm 0.0006$ | - | - |
| | SAGE | $0.0626 \pm 0.0023$ | $\mathbf{0.0606} \pm 0.0016$ | $0.0628 \pm 0.0026$ | $\mathbf{0.0606} \pm 0.0011$ |
| | GAT | $0.0637 \pm 0.0005$ | $\mathbf{0.0606} \pm 0.0016$ | - | - |
| | GTR | $0.0606 \pm 0.0011$ | $0.0560 \pm 0.0010$ | $0.0574 \pm 0.0012$ | $\mathbf{0.0554} \pm 0.0013$ |
| | GIN | $0.0528 \pm 0.0011$ | $0.0481 \pm 0.0007$ | $0.0470 \pm 0.0006$ | $\mathbf{0.0468} \pm 0.0006$ |
| | MGN | $0.0463 \pm 0.0005$ | $0.0447 \pm 0.0009$ | $0.0448 \pm 0.0011$ | $\mathbf{0.0432} \pm 0.0003$ |

### I.10. Ablation Study on Core Components

To disentangle the individual contributions of complementary learning (CL) and inductive biases (IB), we evaluate the performance of all four combinations (no CL + no IB, CL only, IB only, and CL + IB) under the same data configuration (*i.e.*, $N_h = 20$, $N = 200$), while also reporting the standard fully supervised baseline trained only on 20 paired samples ($N_h = N = 20$) for reference. For comparison, the fully supervised baseline is assumed to use all 200 paired HR–LR samples (*i.e.*, $N_h = N = 200$).

*Table 31.* Ablation study on complementary learning (CL) and inductive biases (IB). All experiments use the MGN-based SuperMeshNet and Dataset 1.

| Methods | IB | CL | $N_h$ | $N$ | RMSE |
|---|---|---|---|---|---|
| No CL + no IB (20,20) | X | X | 20 | 20 | 0.0655 |
| No CL + no IB (20, 200) | X | X | 20 | 200 | 0.0454 |
| CL only | X | O | 20 | 200 | 0.0269 |
| IB only | O | X | 20 | 200 | 0.0371 |
| CL + IB (SuperMeshNet) | O | O | 20 | 200 | 0.0226 |
| Baseline: fully supervised | X | X | 200 | 200 | 0.0228 |

The results in Table 31 demonstrate that:

- both CL and IB indeed contribute to improved performance relative to the case of no CL + no IB (20, 200).

- Both components are necessary for SuperMeshNet to outperform the fully supervised baseline trained with 180 additional HR samples.

In particular, in regard to the settings where CL is not used ((i) no CL + no IB (20, 200) and (ii) IB only), we pretrain models to reconstruct LR data from noised LR data (via denoising reconstruction), and fine-tune the models on the 20 paired LR–HR samples. This approach provides a stronger control than the case of no CL + no IB (20, 20) involving training only on the 20 paired samples, as reflected in the lower RMSE.

## I.11. Application to Real-World Geometry and Time-Dependent PDE.

### I.11.1. QUALITATIVE ANALYSIS OF RESULTS ON TIME-DEPENDENT PDE DATASET 2

The visualization in Figure 30 shows that SuperMeshNet successfully predicts the HR counterpart even when the HR data differ substantially from the LR input, whereas conventional fully supervised learning fails despite having access to more HR data. Due to the relatively large grid size of the time-dependent PDE dataset 2, we increased the hidden dimension to 150, whereas the default setting is 30. In addition, because the dataset is defined on a regular grid, we used standard bilinear interpolation instead of $k$NN interpolation.

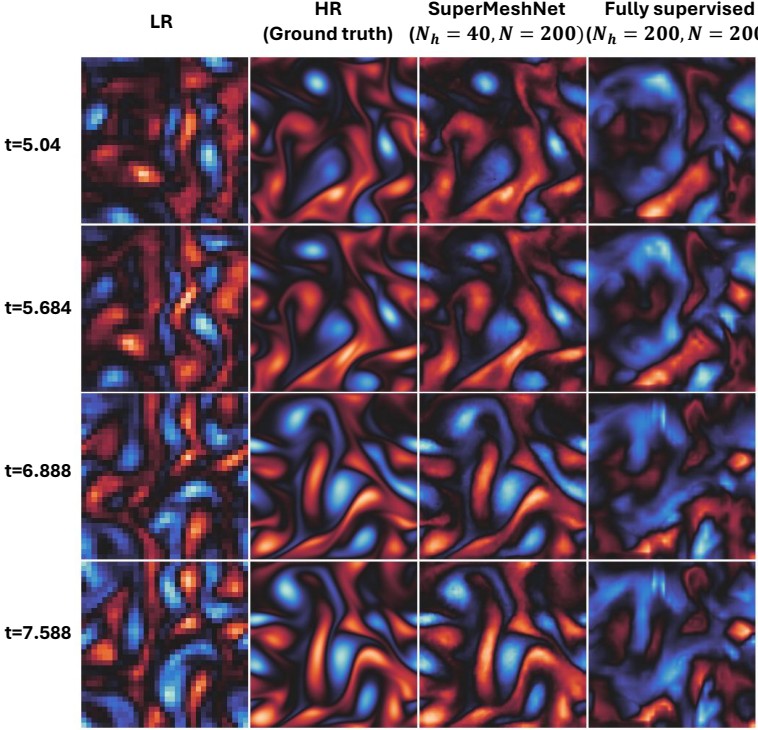

*Figure 30.* LR input, ground truth HR data, and two vorticity predictions produced by SuperMeshNet and full supervision for the time-dependent PDE dataset 2 over multiple timestamps. Here, $N_h$ and $N$ represent the number of HR and LR data samples, respectively. For all cases, MGN is utilized as the underlying MPNN.

### I.11.2. QUANTITATIVE ANALYSIS

We further quantitatively evaluate the applicability of SuperMeshNet to CFD datasets, including one real-world geometry dataset and two time-dependent PDE datasets. The results in Table 32 show that SuperMeshNet, trained with only 40 HR data samples, achieves even a lower error than the case of the fully supervised model trained with 200 HR data samples, demonstrating its effectiveness on real-world geometry and time-dependent PDE datasets.

*Table 32.* The RMSE of MGN-based SuperMeshNet trained with $N_h = 40$ HR data samples and $N = 200$ LR data samples for the real-world geometry and time-dependent PDE datasets, in comparison with fully supervised MGNs trained with $N_h = N = 200$. The best-performing result in each case is highlighted in **bold**.

| Dataset | Methods | $N_h$ | $N$ | RMSE |
|---|---|---|---|---|
| Real-world geometry | SuperMeshNet | 40 | 200 | **0.0559** |
| | fully supervised | 200 | 200 | 0.0584 |
| Time-dependent PDE 1 | SuperMeshNet | 40 | 200 | **0.0700** |
| | fully supervised | 200 | 200 | 0.0793 |
| Time-dependent PDE 2 | SuperMeshNet | 40 | 200 | **0.0543** |
| | fully supervised | 200 | 200 | 0.1613 |

## I.12. Analysis on HR Data Sampling Strategies

In this section, we empirically analyze how HR data sampling strategies affect performance in terms of RMSE.

### I.12.1. IMPACT OF DIFFERENT 20 HR SUBSETS

Table 33 shows that using different subsets of 20 HR samples (*i.e.*, $N_h = 20$) introduces a moderate variance: the RMSE standard deviation (STD) is about 6.7% of the mean RMSE and is roughly twice as large as in the case of using the same 20 HR samples. In the same-samples setting, all five training runs use the same 20 HR samples drawn once from a uniform distribution, whereas, in the different-samples setting, each run uses a distinct set of 20 HR samples independently drawn from the same distribution. For both settings, we measure the RMSE from each training run and report the mean and STD across the five runs. These results indicate that, under a fixed sampling strategy, the variability induced by using different subsets of 20 HR samples remains moderate.

*Table 33.* Effect of different HR subsets on the mean and STD of RMSE for MGN-based SuperMeshNet trained with $N_h = 20$ and $N = 200$ on Dataset 1.

| Strategy | RMSE mean | RMSE STD |
|---|---|---|
| Same random samples | 0.0226 | 0.0007 |
| Different random samples | 0.0208 | 0.0014 |

### I.12.2. IMPACT OF DISTRIBUTION MISMATCH

We further examine how the distributional mismatch between two cases using all 200 LR samples and the 20 LR samples corresponding to selected HR samples affects performance. The mismatch is quantified using maximum mean discrepancy (MMD). As summarized in Table 34, MMD strongly correlates with RMSE: subsets that better match the overall LR distribution yield lower RMSE.

To study this, we compare three sampling strategies. MMD-minimizing greedy sampling (kernel herding (Chen et al., 2010)) begins by randomly selecting one HR sample from the pool of 200 candidates and then iteratively chooses the next HR sample whose corresponding LR counterpart most reduces the MMD between the selected subset and the full LR distribution. This procedure ensures that the selected HR samples closely represent the overall dataset. In contrast, MMD-maximizing greedy sampling intentionally selects the next point that most increases the MMD, pushing the subset away from the full distribution. As demonstrated below, the selected sampling strategy has a significant impact on the RMSE of SuperMeshNet.

*Table 34.* Impact of distribution mismatch-based HR data selection strategies on RMSE for MGN-based SuperMeshNet trained with $N_h = 20$ and $N = 200$ on Dataset 1.

| Strategy | RMSE | MMD |
|---|---|---|
| Same random samples | 0.0226 | 0.147 |
| MMD-minimizing greedy sampling | 0.0185 | 0.058 |
| MMD-maximizing greedy sampling | 0.0259 | 0.330 |

### I.12.3. IMPACT OF ACTIVE LEARNING AND HYBRID STRATEGY

We investigate an inconsistency-based active learning strategy that selects HR samples based on the discrepancy between pseudo-labels generated by the main model $F_\theta$ and the auxiliary model $G_\phi$. Training is initialized with 10 HR samples selected using an MMD-minimizing strategy at the first epoch. Subsequently, one HR sample is added at each epoch by selecting the sample with the highest inconsistency loss, until the total number of HR samples reaches 20. We further evaluate a hybrid strategy that combines the inconsistency-based active learning with MMD-based selection. Specifically, when selecting a next HR sample during active learning, the top 10 candidates are first selected to minimize MMD, and the most inconsistent sample is then chosen from this subset.

As shown in Table 35, inconsistency-based active learning improves performance over random sampling. Moreover, the hybrid strategy achieves the lowest RMSE, indicating that combining inconsistency-based selection with distribution-aware sampling leads to more effective HR data acquisition.

*Table 35.* Impact of inconsistency-based active learning and MMD-based HR sample selection on RMSE for MGN-based SuperMeshNet, trained with $N_h = 20$ and $N = 200$ on Dataset 1.

| Strategy | RMSE |
|---|---|
| Same random samples | 0.0226 |
| MMD-minimizing greedy sampling | 0.0185 |
| Inconsistency-based active learning | 0.0206 |
| Hybrid(inconsistency-based active learning + MMD-minimizing sampling) | 0.0174 |

### I.13. Training Stability

To assess training stability, we plot the loss curves over epochs for five different random seeds. As shown in Figure 31, the loss consistently decreases in all runs without exhibiting divergence.

Although a pseudo-label generated by one model may contain potential errors, such errors do not lead to catastrophic mutual reinforcement. This is because every loss computation judiciously incorporates both pseudo-label-based mutual supervision and ground truth–based supervision. The presence of true labels at every iteration constrains error propagation, preventing any amplification caused by imperfect pseudo-labels. Consequently, complementary learning in our SuperMeshNet framework maintains stable optimization dynamics.

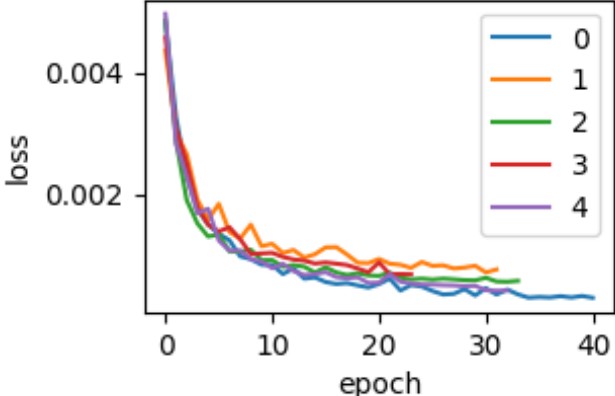

*Figure 31.* Loss curves for five different random seeds, each shown in a different color. For all cases, the MGN-based SuperMeshNet is trained using $N_h = 20$ and $N = 200$ from Dataset 1.

To further investigate training stability, we conduct controlled experiments by injecting errors into pseudo-labels. As illustrated in Figure 32(a), error amplification emerges once the pseudo-label error exceeds a threshold of 0.02. This phenomenon becomes substantially more pronounced when training relies solely on unsupervised loss without access to true labels, as shown in Figure 32(b). These observations highlight the critical role of supervised loss in stabilizing training, as it anchors predictions to ground truth at each iteration and mitigates uncontrolled error propagation.

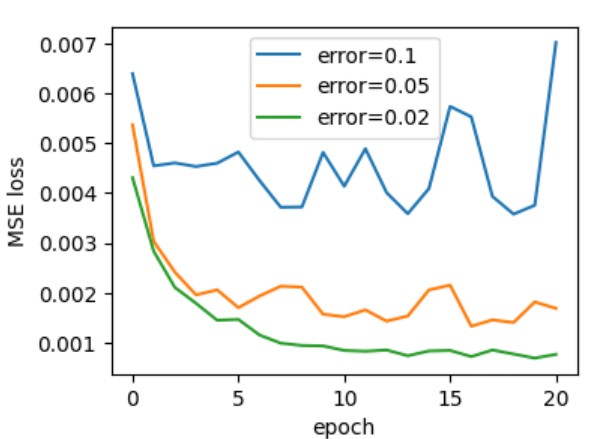

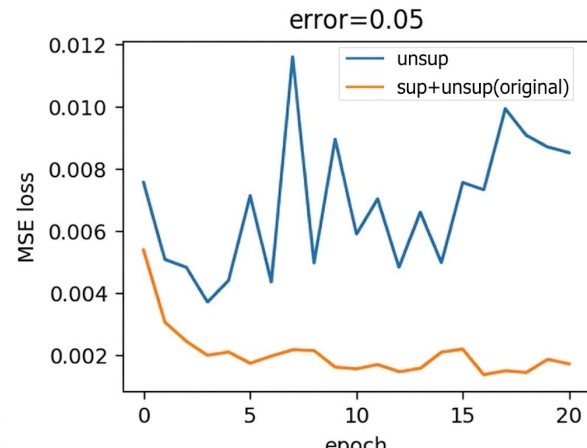

*(a)* Effect of pseudo-label error magnitude on training stability.    *(b)* Effect of supervised loss on training stability.

*Figure 32.* Stability analysis via controlled pseudo-label perturbations. For all cases, the MGN-based SuperMeshNet is trained using $N_h = 20$ and $N = 200$ from Dataset 1.

## I.14. Analysis on $k$NN Interpolation

### I.14.1. RELATIVE COMPUTATIONAL COST OF $k$NN INTERPOLATION

We acknowledge that $k$NN interpolation introduces additional overhead, particularly for very small mesh sizes. As presented in Table 36, the relative interpolation cost indeed grows as the task difficulty increases. However, even after including $k$NN interpolation, the total inference cost remains far lower than the cost of running the HR simulation for all mesh sizes, as indicated in Figure 29. Thus, while $k$NN interpolation may act as a localized bottleneck (particularly for very small mesh sizes), it does not diminish the substantial overall speed-up achieved by SuperMeshNet. We note that our framework is not tied to $k$NN and can leverage more efficient interpolation (*e.g.*, bilinear interpolation) on structured grid datasets. Developing a lighter interpolation method for irregular mesh datasets remains an important direction for future work.

*Table 36.* Relative overhead of $k$NN interpolation across different mesh sizes. All experiments are conducted using the MGN-based SuperMeshNet trained with $N_h = 20$ and $N = 200$ on Dataset 1.

| Mesh size | ($k$NN interpolation time) / (total inference time) |
|:---:|:---:|
| 8 | 17.0% |
| 4 | 18.6% |
| 2 | 29.4% |
| 1 | 71.2% |

I.14.2. SINGULAR POINT ANALYSIS

To evaluate the effect of geometric singularity on super-resolution performance, we additionally generate a dataset that explicitly includes a singular point. Representative HR and LR samples from this dataset are shown in Figure 33.

The computational domain is defined by six vertices: (0, -0.25), (-0.25, -0.25), (-0.25, 0.25), (0.25, 0.25), (0.25,0), and $(x_0, y_0)$, where $(x_0, y_0)$ denotes the location of the singular point. Both $x_0$ and $y_0$ are independently sampled from a uniform distribution on [-0.125, 0.125]. The default mesh sizes for LR and HR meshes are set to 0.04 and 0.01, respectively. However, within the vicinity of the singular point, the mesh is refined by a factor of four for each resolution.

On this domain, the Poisson equation below is solved :

$$\nabla^2 u = ((x - x_0)^2 + (y - y_0)^2)^{-1}, \tag{27}$$

where $u$ denotes the electrical potential. The boundary condition is fixed at 0. After solving the equation, the magnitude of the electric field is calculated at each node of the mesh. Examples of LR and HR fields are visualized in Figure 33.

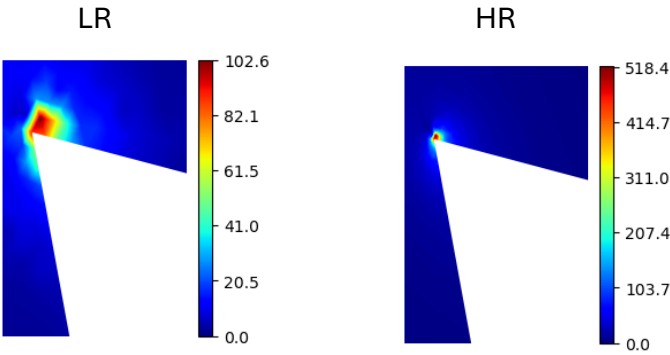

*Figure 33.* Examples of LR and HR data samples from the dataset including a singular point.

We compare the HR fields predicted by SuperMeshNet with those obtained via pure $k$NN interpolation without neural network-based prediction in Figure 34. Both methods exhibit relatively high errors at the singular point. Nevertheless, the MPNN in SuperMeshNet noticeably mitigates the severity of these errors: both the magnitude and spatial extent of the high error region are reduced compared to the case of pure interpolation. Specifically, the maximum error near the singularity is 4.18 for pure $k$NN interpolation, whereas SuperMeshNet reduces this peak error to 1.57.

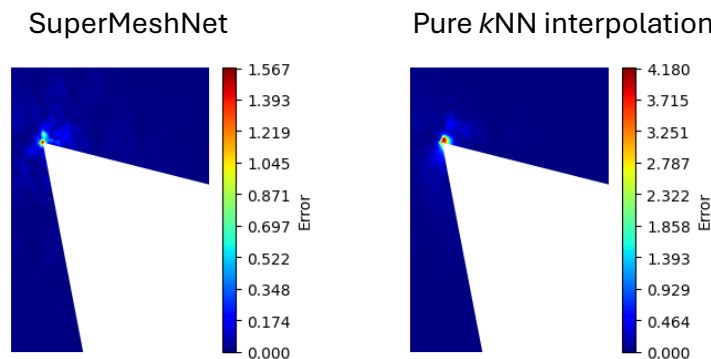

*Figure 34.* Comparison of predictions by SuperMeshNet and pure $k$NN interpolation.

## I.15. Comparison with a PINN

As summarized in Table 37, we compare a physics-informed neural network (PINN) (Raissi et al., 2019) and SuperMeshNet in terms of computational time and RMSE performance.

First, we describe the experimental setting. We conduct evaluations on the Poisson equation:

$$\nabla^2 u = \sqrt{x^2 + y^2},$$

where $u$ is the solution and $(x, y)$ denotes the spatial coordinate. The computational domain is an L-shaped region with six vertices:

$$(0, -0.25), \ (-0.25, -0.25), \ (-0.25, 0.25), \ (0.25, 0.25), \ (0.25, 0), \ (0, 0),$$

and homogeneous Dirichlet boundary conditions ($u = 0$) are applied on all boundaries.

For training SuperMeshNet, we construct a dataset consisting of 20 HR and 200 LR samples. The domain is the same L-shape, except that the final vertex is replaced by a variable point $(x_0, y_0)$, where both $x_0$ and $y_0$ are independently sampled from a uniform distribution on $[-0.125, 0.125]$. The default sizes for LR and HR meshes are set to 0.04 and 0.01, respectively; however, within the vicinity of the singular point, the mesh is refined by a factor of four for each resolution. On this domain, to generate LR and HR data, we solve the following Poisson equation:

$$\nabla^2 u = \sqrt{(x - x_0)^2 + (y - y_0)^2}.$$

Next, we show experimental results, which exhibit a clear trade-off: SuperMeshNet achieves substantially lower RMSE, while PINN attains much lower total computation time per instance.

However, these results are not conclusive. Once SuperMeshNet is trained, it can be applied to any new instance without retraining, whereas a PINN must be retrained for every new instance. Thus, when a large number of parameterized PDE instances must be solved, the initial training cost of SuperMeshNet becomes amortized. Likewise, the RMSE of a PINN can potentially be improved by recent advanced optimization techniques. A more extensive comparison is left for future work.

*Table 37.* Comparison between PINN and MGN-based SuperMeshNet.

| Methods | Data preparation time + training time (s) | Inference time (s) | RMSE |
|---|---|---|---|
| PINN | 5.8 | 0.0003 | $8.0 \times 10^{-4}$ |
| SuperMeshNet | 16467.3 | 0.0061 | $1.0 \times 10^{-4}$ |

