# OpenReview forum: "Semi-Supervised Neural Super-Resolution for Mesh-Based Simulations"
_ICML.cc/2026/Conference — ICML 2026 regular_

### Official Review · Reviewer_2ccX · 2026-03-10

**Soundness:** 3
**Presentation:** 3
**Significance:** 2
**Originality:** 2
**Overall Recommendation:** 4
**Confidence:** 5

**Summary:**

This paper proposes SuperMeshNet, a semi-supervised super-resolution framework for mesh-based PDE simulations that aims to reduce the need for expensive HR supervision. The key technical idea is complementary learning: jointly training (i) a primary MPNN mapping LR→HR and (ii) an auxiliary MPNN predicting HR–HR differences between two LR inputs with kNN interpolation to handle mesh mismatch. The two models generate pseudo-targets for each other to exploit abundant unpaired LR samples. The method is further augmented with inductive biases, node-level and message-level centering, claimed to smooth optimization and improve performance across multiple MPNN architectures.

**Compliance With Llm Reviewing Policy:**

Affirmed.

**Final Justification:**

The authors have re-evaluated their models with more physics parameters, like lift and drag coefficients. I'm satisfied with their rebuttals.

**Key Questions For Authors:**

1. For FEM datasets and CFD time-dependent PDE 1, their HR nodes contains just few thousands of nodes. Are they suitable to be considered as HR ?
2. Can you report at least one physics-aware metric for the CFD cases?
3. The auxiliary model trains on two HRs only makes sense for time-dependent datasets, as the two HRs are related, assuming they are coming from the same problem and setup, but at different timesteps. What is the rationale behind this prediction of a difference between two HR solutions, especially if these two HR solutions are from steady-state FEM datasets with different setups/parameters? Basically, these two HRs are totally different physical solutions, just solved under the same physical laws.

**Limitations:**

yes

**Strengths And Weaknesses:**

Strengths
1. Results are reported across six representative MPNNs (GCN/SAGE/GAT/GTR/GIN/MGN), supporting the claim of MPNN-agnostic applicability.
2. Well written with clear illustrations.

Weaknesses
1. There is no validation for datasets generated by authors. In CFD or any numerical simulation fields, validation of the numerical solution with experimental data or conduction of mesh convergence study is the very first step to justify your solutions, else “garbage in, garbage out”. Although this is an AI conference, the physics should still be upheld, since the authors submitted to the Applications-physics track; the physics must be valid and properly studied, otherwise this is meaningless.
2. Missing standard deviation reporting
3. Evaluation mostly emphasizes RMSE; limited physics-aware validation. Since this is a physics track, at least some physics-relevant measures (e.g., constraint residuals, conservation error, lift and drag coefficients) beyond field RMSE, especially for CFD/time-dependent cases. Right now, the paper’s main metric is RMSE. The model should not only minimize the fields but also ensure the outputs obey the underlying physics, otherwise this seems like an image processing tool.

---

> ### Author Rebuttal · Authors · 2026-03-30
>
> Dear reviewer 2ccX,
>
> Thank you for your valuable feedback highlighting the importance of physical validity. We understand that  "physics in, physics out" is paramount.
> # Figures
> Please note that the figures referenced in our response are available at [https://anonymous.4open.science/r/author](https://anonymous.4open.science/r/author).
> # W1. Validation of datasets
> - FEM datasets: We have performed mesh convergence tests (Fig. R7). The results show consistent error reduction with mesh refinement, indicating convergence toward a mesh-independent solution and the sufficiently resolved numerical solutions.
> - Revision of the time-dependent PDE dataset 1: During validation, we observed that the original dataset exhibited unstable flow behavior. We have therefore revised the setup (inlet condition and symmetry breaking) accordingly. The updated dataset shows more stable wake formation and consistent 1) enstrophy, 2) drag, and 3) lift coefficient evolution (Fig. R9). We have used only data after the transient regime (t>5s) for training. After reconstructing the dataset with physically consistent settings, we have re-run experiments. The table below and Fig. R10 demonstrate that SuperMeshNet continues to outperform the fully supervised baseline while using significantly fewer HR samples, confirming that our main findings remain robust.
> |Method|RMSE|
> |--|--|
> |SuperMeshNet ($N_h$=20 $N$=200)|0.0049|
> |Fully supervised ($N_h$=$N$=200)|0.0069|
> - Real-world geometry dataset: Mean divergence (normalized with speed/length) correspond to 0.0047 and 0.0071 for HR and LR simulations, respectively, indicating small mass conservation error.
> - Time-dependent PDE dataset 2: Enstrophy stably decreases, and divergence (normalized with speed/length) remains extremely small (<~1e-12) for both LR and HR simulations (Fig. R8), indicating numerically consistent solutions.
>
> We shall add the aforementioned analysis into the camera-ready version upon acceptance.
> # W2. Standard deviations
> Please note that full results including standard deviations are available in Appendices I, K, and M.
> # W3 & Q2. Physics-aware evaluation
> We have evaluated enstrophy evolution over time.
> - Time-dependent PDE 1 dataset: As shown in Fig. R11, SuperMeshNet accurately captures the oscillatory behavior, with improved amplitude and phase alignment compared to the fully supervised baseline, even with fewer HR samples.
> - Time-dependent PDE 2 dataset: As shown in Fig. R12, SuperMeshNet better tracks the enstrophy decay rate, while the fully supervised baseline exhibits overly dissipative behavior.
> - We believe incorporating explicit physical constraints (e.g., physics-informed losses) could further improve physical consistency.
>
> We will include these results in the camera-ready version upon acceptance.
> # Q1. HR dataset scale
> We agree that a few thousand nodes are smaller than those used in large-scale CFD or FEM simulations and are not sufficient to fully assess scalability. We consider our two CFD datasets, with \~50k and \~1M nodes, to be relatively large compared to prior work [1]. While super-resolution difficulty is primarily governed by the relative resolution ratio rather than the absolute scale, we further examine scalability by conducting additional experiments on large-scale regular grid datasets (\~2M nodes) from BLASTNet 2.0 [2]:
>
> | Sub-dataset | Fully supervised ($N_h$=$N$=50) | SuperMeshNet ($N_h$=10 $N$=50) |
> |--|--|--|
> | Forced hit | 0.0451 | 0.0230 |
> | Parametric variation | 0.0785 | 0.0643 |
>
> SuperMeshNet still outperforms the fully supervised baseline while using fewer HR samples, implying that our findings generalize across different dataset scales. Despite these gains, creating larger-scale irregular mesh-based datasets is an important direction for future work.
>
> [1] W. Xu et al., CPS-IoT Week’23
>
> [2] W. Chung et al, NeurIPS 2023
>
> # Q3. Rationale for the auxiliary model
> The two HR solutions are not arbitrary, even in steady-state FEM datasets; they are governed by the same PDE and differ only in a single parameter (e.g., force angle or geometry). Therefore, their difference reflects the physical response of the system to a parameter change. For well-posed problems, such responses are smooth and structured, meaning that the change in solutions is not random but follows consistent physical behavior induced by the governing equations. This is the case, for our datasets, where parameter variations lead to stable and predictable solution changes. In this sense, the auxiliary model learns how the solution varies under parameter perturbations, which is a physically meaningful quantity rather than an arbitrary difference. Thus, the auxiliary model guides the primary model to produce predictions that follow the physical response of the system to parameter variations.
> However, we note that this assumption may weaken in regimes with strong nonlinearities or bifurcations. We will provide this rationale in the manuscript upon acceptance.

---

> > ### Author Rebuttal · Reviewer_2ccX · 2026-04-01
> >
> > 1. The convergence plots clearly showed that your datasets are still far from convergence, the L2 error decreases as the mesh is refined, and there is no sign of convergence. Also, when conducting a mesh convergence study, we examine key results that indicate a plateauing trend with mesh refinement. For instance, the authors should examine the maximum stress and strain concentrated around the holes for FEM datasets, lift & drag coefficients for CFD datasets, not the L2 error over the domain. Still, if looking at the L2 error, it should show a plateau with mesh refinement, not a diagonal line.
> >
> > 2. For physics-aware evaluation, could the authors provide the stress/strain evaluation (compared with HR) around the holes on FEM datasets as well? Also, since time-dependent PDE 1 and Real-World Geometry datasets are of the same nature (flow over an obstacle), the very key results are the lift and drag coefficients of the cylinder and motorbike, why the authors did not consider evaluating them instead of looking at enstrophy? I would like to see how the performance of your model is on the prediction of both lift and drag coefficients.
> >
> > 3. I'm satisfied with the explanation of both the HR scale and the use of the auxiliary model, which utilizes the difference between two HR solutions.
> >
> > ##################
> >
> > **Reply to authors' rebuttal comments**
> >
> > Appreciate the results on both mesh convergence and physics-related evaluations. I will increase the score accordingly.

---

> > > ### Author Response · Authors · 2026-04-07
> > >
> > > Dear reviewer 2ccX,
> > >
> > > Thank you for your careful follow-up and clear guidance. We have revised our analysis to directly address your concerns by focusing on physically meaningful quantities and explicitly demonstrating convergence behavior.
> > >
> > > **Figures**: [https://anonymous.4open.science/r/author](https://anonymous.4open.science/r/author).
> > >
> > > # Mesh convergence
> > > ## FEM datasets
> > > -	Stress/electric field convergence: We have examined stress/electric field in high-concentration regions. As shown in Fig. R14, these quantities exhibit clear saturation as the mesh is refined, indicating convergence toward mesh-independent regimes.
> > >
> > > -	L2 error: The original graph appears indeed linear because both axes in Fig. R7 were plotted in log scale to illustrate exponential convergence. We have redrawn the L2 error in linear scale in Fig. R13, which clearly shows that the error approaches a plateau.
> > >
> > > -	HR dataset scale: Our HR FEM datasets contain ~4000 nodes, which lie near the onset of the convergence plateau. While further refinement would slightly improve accuracy, the gain is marginal compared to the substantial increase in data generation cost. Since our objective is to study data-efficient super-resolution, we believe using HR data in this regime is acceptable. Moreover, if the HR solutions were fully mesh-independent, the super-resolution task itself would become trivial.
> > >
> > > ## Cylinder dataset (time-dependent PDE 1 dataset)
> > > -	Drag and lift coefficients: We have analyzed mean drag, drag amplitude, and lift amplitude, which reflect vortex shedding. These quantities approach constant values with mesh refinement (Fig. R15), indicating convergence.
> > >
> > > -	Dataset revision: We have reduced time step and slightly increased viscosity to improve numerical stability without altering key flow characteristics (e.g., vortex shedding behavior). We have also revised the dataset to include ~20,000 nodes so that the HR data fall near the onset of the plateau regime, consistent with the FEM datasets.
> > >
> > > ## Motorbike dataset (Real-world geometry dataset)
> > > -	Drag/lift coefficients convergence: Both drag and lift coefficients show converging trends with mesh refinement (Fig. R16).
> > >
> > >
> > > -	Dataset revision: We have revised the dataset to include ~80,000 nodes such that the dataset operates near the onset of the convergence plateau.
> > >
> > > # Physics-aware evaluation
> > > ## FEM datasets
> > > SuperMeshNet predicts stress/electric field in concentrated regions with <5% error, while outperforming the fully supervised baseline with fewer HR samples.
> > >
> > > ### Dataset 1
> > >
> > > | |Stress|Relative error|
> > > |-|-|-|
> > > |HR ground truth|1.65e8| |
> > > |SuperMeshNet($N$=40, $N_h$=200)|1.70e8|0.0255|
> > > |Fully supervised ($N=N_h$=200)|1.72e8|0.0382|
> > >
> > > ### Dataset 2
> > > | |Stress|Relative error|
> > > |-|-|-|
> > > |HR ground truth|2.00e8| |
> > > |SuperMeshNet($N$=40, $N_h$=200)|1.98e8|0.0100|
> > > |Fully supervised ($N=N_h$=200)|1.95e8|0.0223|
> > >
> > > ### Dataset 3
> > > | |Electric field|Relative error|
> > > |-|-|-|
> > > |HR ground truth|4.74e1| |
> > > |SuperMeshNet( $N$=40, $N_h$ =200)|4.54e1|0.0429|
> > > |Fully supervised ($N=N_h$=200)|4.38e1|0.0770|
> > >
> > > ## Cylinder dataset (time-dependent PDE 1 dataset)
> > > SuperMeshNet predicts drag/lift metrics with <0.5% error, while outperforming the fully supervised baseline (Fig. R18).
> > >
> > > | |drag coefficient: mean (relative error)|drag coefficient: amplitude (relative error)|lift coefficient: amplitude (relative error)|
> > > |-|-|-|-|
> > > |HR ground truth|1.39|0.0451|0.698|
> > > |SuperMeshNet($N$=40, $N_h$=200)|1.40(0.005)|0.0453(0.003)|0.696(0.003)|
> > > |Fully supervised ($N=N_h$=200)|1.40(0.007)|0.0455(0.008)|0.691(0.010)|
> > >
> > >
> > >
> > > ## Motorbike dataset (Real-world geometry dataset)
> > > The following table shows that SuperMeshNet achieves a low relative error compared to that of the fully supervised baseline in drag prediction despite using much fewer HR samples. In contrast, SuperMeshNet exhibits a modestly larger relative error in lift coefficient, which can be attributed to its small magnitude, for which small absolute deviations can result in large relative errors.
> > >
> > > | |drag coefficient(relative error)|lift coefficient(relative error)|
> > > |-|-|-|
> > > |HR ground truth|0.3724|0.0368|
> > > |SuperMeshNet($N$=40, $N_h$=200)|0.3778 (0.014)|0.0433(0.177)|
> > > |Fully supervised ($N=N_h$=200)|0.3653(0.019)|0.0380(0.033)|
> > >
> > > ## Weighted loss
> > > We have modified the original loss to a weighted MSE loss to account for the scale difference between velocity (u) and pressure (p) for the cylinder and motorbike datasets. Since the magnitudes of u and p differ significantly, directly using a uniform loss leads to imbalance where one variable dominates the optimization, which can bias the optimization toward one variable. To mitigate this, we have applied channel-wise weighting to balance their contributions during training.
> > >
> > >
> > > We will include these physics-aware evaluations in the main text of the camera-ready version upon acceptance. We believe these additional analyses directly address your concerns on physical validity and significantly strengthen the credibility of our results.

---

### Official Review · Reviewer_a6jp · 2026-03-10

**Soundness:** 3
**Presentation:** 3
**Significance:** 3
**Originality:** 3
**Overall Recommendation:** 4
**Confidence:** 4

**Summary:**

The authors propose SuperMeshNet, a highly data-efficient framework that introduces a novel semi-supervised "complementary learning" strategy and two simple yet highly effective inductive biases (node- and message-level centering). This approach is evaluated with various datasets, outperforming other model architectures.

**Compliance With Llm Reviewing Policy:**

Affirmed.

**Final Justification:**

I appreciate the authors' response. I would keep my assessment unchanged.

**Key Questions For Authors:**

1. The author's explanation of why complementary learning is better than fully supervised learning is not specific to the super-resolution of mesh-based simulations. Does this mean we could leverage the proposed semi-supervised learning paradigm to imporve all super-resolution tasks?

**Limitations:**

yes

**Strengths And Weaknesses:**

Strengths:
1. The paper is well-structured and clearly written.
2. The proposed complementary learning paradigm proves to be robust and highly effective, leveraging the unpaired LR data to great advantage.
3. The two inductive biases (node- and message-level centering) introduced in this work prove to be exceptionally effective and well-justified.

Weaknesses:
1. The training samples used in the time-dependent experiments seem to be distinguished by time steps instead of the specific parameter $\mu$ that defines different PDE variations. Consequently, this raises potential concerns about the validity of the i.i.d. assumption for the training dataset.

---

> ### Author Rebuttal · Authors · 2026-03-30
>
> Dear Reviewer a6jp,
>
> Thank you for your positive feedback and your recognition of the strengths of our work.
> # 1. i.i.d. assumption for time-dependent PDE datasets
> We agree that samples from the same trajectory are temporally correlated and thus are not strictly i.i.d.. However, our problem setting and methodology are not built upon the i.i.d assumption. In addition, our goal in the time-dependent setting is to evaluate whether the proposed framework is applicable to time-dependent PDE datasets rather than generalization across different initial conditions. Please note that constructing training datasets using multiple time steps from the same trajectory is a common and practical setup, as also adopted in prior work [1,2].
>
> [1] Z. Li et al., ICLR 2021, Fourier Neural Operator for Parametric Partial Differential Equations
>
> [2] T. Pfaff et al., ICLR 2021, Learning Mesh-Based Simulation with Graph Networks
>
> # 2. Applicability to other super-resolution tasks
> Yes, the proposed complementary learning is not inherently restricted to mesh-based simulations. At its core, the method leverages unpaired LR data through mutual supervision between two complementary models, which can be naturally extended to other super-resolution tasks such as image super-resolution. That being said, we would like to emphasize simulation domains, where data scarcity, corresponding to our target problem setting, is significantly more severe compared to the case of image super-resolution.
>
> We will incorporate this discussion into the camera-ready version once our paper is accepted.

---

> > ### Author Rebuttal · Reviewer_a6jp · 2026-04-03
> >
> > Thanks for the detailed rebuttal. I appreciate the clarifications, but my overall assessment remains unchanged.

---

> > > ### Author Response · Authors · 2026-04-07
> > >
> > > Dear reviewer a6jp,
> > >
> > > We sincerely thank you for your positive assessment and constructive feedback. We are glad that our responses helped clarify the key aspects of our work and have fully addressed your concerns. Thank you again for your time and support.

---

### Official Review · Reviewer_isWS · 2026-03-12

**Soundness:** 3
**Presentation:** 3
**Significance:** 3
**Originality:** 3
**Overall Recommendation:** 5
**Confidence:** 3

**Summary:**

This paper introduces a new semi-supervised learning framework, _complementary learning_, for super-resolution with reduced HR supervision. It also proposes new normalization-like operations for MPNNs, _node-level centering_ and _message-level centering_, to improve accuracy. The method is evaluated across multiple datasets and MPNN architectures, and the results show that even with 90% fewer HR samples, the proposed approach is competitive with fully supervised baselines in many settings and outperforms them in some cases.

**Compliance With Llm Reviewing Policy:**

Affirmed.

**Final Justification:**

My main concerns were the training stability and the impact of the feature extractor. For training stability, the authors provided additional experiments clarifying possible conditions under which training becomes unstable and showed that supervised signals can effectively mitigate it. For the feature extractor, their explanation of the accuracy–efficiency trade-off in terms of pseudo-label diversity clearly resolved my concerns. Therefore, I have increased my final score from 4 to 5.

**Key Questions For Authors:**

- **Q1**: How stable is the proposed complementary learning framework in practice? In particular, could differences in the learning progress of the HR predictor $F$ and the HR-difference predictor $G$ lead to a negative feedback loop, where a relatively poor $G$ degrades the training targets for $F$, which in turn further worsens overall performance (i.e., catastrophic mutual reinforcement)? Appendix R provides numerical evidence, but could the authors clarify more explicitly, either empirically or theoretically, under what general conditions error amplification is unlikely to occur? If this point is regarded as future work, it should be stated explicitly as a limitation in the main text.
- **Q2** (L266-270): Why is the LR input for the Time-dependent PDE1 dataset generated by downsampling from HR, whereas the other datasets use independently generated LR inputs? This reason should be briefly stated in the main text.
- **Q3** (L419-422 and Fig. 9): In Fig. 9, could the strong performance of the proposed method be due in part to the shared feature extractor? The paper attributes the improvement primarily to introducing $G$. If $F$ and $G$ were independent and $G$ were a perfect predictor, the setting would become closer to a fully supervised one. Is it possible that one contributing factor is that the shared feature extractor learns HR-HR relationships through $G$, which then also benefits $F$?
- **Q4** (Appendix Y): Appendix Y states that using separate feature extractors for $F$ and $G$ improves accuracy (L2809), but Table 31 appears inconsistent with this statement: the RMSE for the separate feature extractor setting (0.0226) is higher than that for the shared feature extractor setting (0.0192). Could the authors clarify this discrepancy? More generally, is the benefit of sharing the feature extractor task- or dataset-dependent in terms of accuracy?

If Q1, Q3, and Q4 are addressed convincingly, I may revise my overall recommendation from 4 (Weak Accept) to 5 (Accept).

**Limitations:**

W3 is briefly addressed in Appendix R, but is not discussed in the main text. If W3 is regarded as future work, it should be stated explicitly as a limitation in the main text.

**Strengths And Weaknesses:**

**Strengths**
- **S1**: The proposed learning framework is applicable to unstructured meshes and has the potential to substantially reduce the computational cost of generating HR training samples.
- **S2**: The proposed centering operations are simple and do not require additional trainable parameters. They appear effective for super-resolution tasks in which performance does not strongly depend on the global mean of the features.
- **S3**: The method is validated across multiple MPNN architectures and multiple datasets (e.g., electrostatics, fluid flow, and elasticity), suggesting that it may remain effective across problems with different underlying physics.

**Weaknesses**
- **W1**: The proposed centering operations explicitly remove the feature mean, so it is unclear whether they remain effective for tasks beyond super-resolution, especially for tasks that depend strongly on mean information (L230-234 and Appendix T).
- **W2**: The method uses a large number of unpaired LR samples during training, which may increase training time depending on mesh size (Section 3.4 and L426-433).
- **W3**: The stability of the complementary learning framework is discussed in Appendix R, but its stability under more general conditions remains unclear (see Q1).

---

> ### Author Rebuttal · Authors · 2026-03-30
>
> Dear Reviewer isWS,
>
> We sincerely thank you for your constructive feedback and for your recognition of the strengths of our work.
> # Figures
> Please note that the figures referenced in our response are available at [https://anonymous.4open.science/r/author](https://anonymous.4open.science/r/author).
>
> # W1. Effectiveness of centering
> We agree that, in contrast to super-resolution tasks where relative spatial variation is more critical than absolute feature means, centering operations may be less effective for tasks where the global mean carries essential information, as discussed in Appendix T. However, our focus is on improving super-resolution performance, not on addressing general-purpose tasks.
>
> To avoid overgeneralizing the effects of centering, we will clarify this scope explicitly in the main text of the camera-ready version once our paper is accepted.
>
> # W2. Training time
> We acknowledge that complementary learning increases training time to some extent. However, as discussed in Sections 3.4 (Figure 6) and 4, the overall computational cost can be reduced in practice because generating HR data is significantly more expensive than training, especially when the mesh size decreases. Thus, our approach trades increased training time for substantial savings in data generation cost.
>
> We will further emphasize this trade-off in the camera-ready version once our paper is accepted.
>
> # W3 & Q1. Stability of complementary learning
> To further investigate stability, we have conducted additional experiments by injecting controlled errors into pseudo-labels. When a pseudo-label error exceeds a threshold of 0.02, we observe error amplification during training ([Fig. R3](https://anonymous.4open.science/r/author/Figure_R3.png)). Additionally, when training relies solely on unsupervised loss not using true labels, this effect becomes significantly more severe ([Fig. R4](https://anonymous.4open.science/r/author/Figure_R4.png)). These results imply that the supervised loss plays a critical role in stabilizing training by anchoring predictions to ground truth at each iteration, thereby preventing uncontrolled error propagation. In practice, we consistently include supervised signals, even when they are available only in modest amounts, which ensures stable training.
>
> We will clarify this stability condition in the camera-ready version once accepted and explicitly state in the main text that a more rigorous theoretical analysis remains future work.
>
> # Q2. Reason for downsampling in the time-dependent PDE dataset 1
> In the time-dependent PDE dataset 1, which exhibits wake formation behind a cylinder, we found that time synchronization between independently simulated LR and HR trajectories is not guaranteed. As illustrated by the enstrophy analysis ([Fig. R5](https://anonymous.4open.science/r/author/Figure_R4.png)), where oscillations correspond to wake dynamics, independently generated LR simulations have shown delayed wake development and, more importantly, a different wake frequency compared to HR simulations. This discrepancy breaks the physical correspondence between LR and HR snapshots at the same time step. To ensure a well-defined super-resolution task with aligned spatiotemporal structures, we have therefore adopted LR input samples generated by downsampling from HR data, following standard practice in prior work [1,2].
>
> We will briefly clarify this rationale in the main text of the camera-ready version upon acceptance.
>
> [1] K. Fukami et al., Jornal of Fluid Mechanics 870 (2019), Super-resolution reconstruction of turbulent flows with machine learning
>
> [2] W. T. Chung et al., NeurIPS 2023, Turbulence in Focus: Benchmarking Scaling Behavior of 3D volumetric Super-Resolution with BlLASTNet 2.0
>
>  # Q3 & Q4. Shared vs. separate feature extractors
> Thank you for pointing out the inconsistency in Table 31. We confirm that using separate encoders yields a lower RMSE (0.0192) but requires longer training time (962.85s), while shared encoders are more efficient (263.72s) at the cost of lower accuracy (0.0226). We will correct Table 31 accordingly.
>
> We have adopted shared encoders as a practical trade-off between accuracy and efficiency. We attribute the accuracy difference to the degree of correlation between representations: separate encoders allow the two models to learn less correlated (more independent) representations, resulting in more diverse pseudo-labels. Such reduced correlation mitigates redundancy and provides complementary supervisory signals, thereby strengthening mutual supervision and improving learning (see Section 3.5 and Appendix L). In contrast, shared encoders induce stronger coupling between the two models, increasing correlation, reducing pseudo-label diversity, and slightly limiting performance gains.
>
> We will clarify this trade-off in the camera-ready version once our paper is accepted.

---

> > ### Author Rebuttal · Reviewer_isWS · 2026-04-03
> >
> > Thank you for your clear and helpful response, which resolved my concerns, and I have therefore increased my score from 4 to 5.

---

> > > ### Author Response · Authors · 2026-04-07
> > >
> > > Dear reviewer isWS,
> > >
> > > We greatly appreciate your insightful and constructive feedback. We are pleased that our responses have helped clarify the key aspects of our work and have fully addressed your concerns. Thank you again for your time and consideration.

---

### Official Review · Reviewer_vpK8 · 2026-03-16

**Soundness:** 2
**Presentation:** 3
**Significance:** 2
**Originality:** 3
**Overall Recommendation:** 4
**Confidence:** 3

**Summary:**

This paper proposes SuperMeshNet, a semi-supervised framework for super-resolution of mesh-based PDE simulations. The goal is to reconstruct high-resolution (HR) simulation fields from low-resolution (LR) inputs while minimizing the amount of expensive HR training data required. The proposed method introduces a complementary learning mechanism that jointly trains two message-passing neural networks (MPNNs): a primary model that predicts HR solutions from LR inputs, and an auxiliary model that predicts differences between HR solutions corresponding to different LR inputs. These models provide mutual supervision using pseudo-targets, enabling the framework to leverage large amounts of unpaired LR data alongside a small amount of paired LR–HR samples.

**Compliance With Llm Reviewing Policy:**

Affirmed.

**Final Justification:**

My concerns are resolved. I am happy to increase the evaluation accordingly.

**Key Questions For Authors:**

- The experiments focus primarily on elasticity, Poisson, and Navier-Stokes datasets. How well does the method generalize to more complex multi-physics problems or highly turbulent regimes?
- Since the method relies on kNN interpolation and message passing, how does performance scale with meshes containing millions of nodes?
- The paper shows that HR data selection can influence performance. Would actively selecting informative HR samples (e.g., active learning) further improve results? If so, how sensitive is the framework to HR sampling strategies?
- The method relies on mutual pseudo-supervision between two models. Under what conditions might error reinforcement occur, and how robust is the method when pseudo-labels are inaccurate?
- Would the authors provide more intuitions on why the proposed method could improve performance?

**Limitations:**

yes

**Strengths And Weaknesses:**

### Strengths
- The paper is well written and organized
- The experiments compare the proposed method against a range of relevant baselines, providing a reasonably thorough empirical assessment of the approach.
- The proposed method reduces the amount of high-resolution (HR) supervision required for mesh super-resolution. Experiments show that the method can achieve comparable performance to fully supervised models while using only a small fraction of HR training data.

### Weaknesses
- Most evaluations are on synthetic FEM/CFD datasets; broader real-world simulation scenarios would strengthen the claims.
- The approach relies heavily on kNN interpolation for mesh alignment, which may introduce computational overhead and potential approximation errors.
- The paper lacks intuition explaining why complementary learning improves performance or when it is expected to work best.

---

> ### Author Rebuttal · Authors · 2026-03-30
>
> Dear Reviewer vpK8,
>
> We sincerely thank you for your thoughtful feedback and constructive questions.
> # Figures
> The figures referenced in our response are available at [https://anonymous.4open.science/r/author](https://anonymous.4open.science/r/author).
> # W1 & Q1. Multi-physics or turbulent regimes
> Due to the lack of irregular mesh-based multi-physics datasets, we have conducted additional experiments on two subsets (Forced Hit and Parametric Variation) of BlastNet 2.0 [1], which include regular grid-based multi-physics (reaction-flow) and turbulent flows.
> |Sub-dataset|Fully supervised ($N_h$=$N$=50)|SuperMeshNet ($N_h$=10 $N$=50)|
> |--|--|--|
> |Forced hit|0.0451|0.0230|
> |Parametric variation|0.0785|0.0643|
>
> SuperMeshNet consistently outperforms the fully supervised baseline while using 80% fewer HR samples. Fig. R6 further demonstrates that SuperMeshNet accurately and apparently reconstructs ground truth HR fields.
> We leave experiments on irregular mesh-based multi-physics or turbulence datasets as due to the lack of benchmark datasets.
>
> [1] W. Chung et al, NeurIPS 2023, Turbulence in Foucus: Benchmarking Scaling Behavior of 3D Volumetric Super-Resolution with BLASTNET 2.0 Data
>  # W2 & Q2. kNN interpolation and scalability
> -	Interpolation error: Although kNN interpolation introduces approximation error, the neural network is trained end-to-end to minimize this error. As shown in Appendix S, SuperMeshNet apparently mitigates interpolation artifacts (e.g., near singular regions) compared to the case of pure kNN interpolation.
> - Computational cost: Appendices P.2 and P.3 demonstrate that, even after including kNN interpolation, the total inference cost remains far lower than the cost of running HR simulations. In other words, kNN interpolation does not diminish the substantial overall speed-up achieved by SuperMeshNet.
> - Scalability: kNN search incurs an O(N log N) cost, but this is typically negligible compared to HR simulation time and can be further reduced using approximate methods. We have empirically validated scalability on BlastNet 2.0 (~2M nodes) (see our response to W1 and Q1). Moreover, we note that our framework is not tied to kNN and can leverage more efficient interpolation (e.g., bilinear interpolation) on structured grids.
> # W3 & Q5. Intuition on complementary learning
> Unlike conventional semi-supervised methods where multiple models generate pseudo-labels for the same prediction target (which often leads to correlated errors and confirmation bias), our framework decomposes learning into two structurally different tasks: inter-resolution mapping (LR ->HR) and intra-resolution difference modeling (LR–LR ->HR–HR).
> Because the two models predict different targets, their errors are inherently less correlated compared to conventional co-training (Appendix B). As a result, the auxiliary task provides complementary supervisory signals rather than redundant ones.
>
> Complementary learning is most effective when the intra-resolution relationships are informative. As the dataset consists of solutions of the same PDE under varying parameters, this is typically the case since most PDEs are well-conditioned, where parameter changes lead to stable rather than chaotic or sharply discontinuous solution variations.
> # Q3. HR data selection
> We have newly tested an active learning strategy that selects HR samples based on pseudo-label inconsistency between the two models. We start with 10 HR samples selected at the first epoch using the MMD-minimizing strategy (Appendix U.2), and then add one HR sample at each epoch by selecting the sample with the highest inconsistency loss until the total number of HR samples reaches 20. We have also tested a hybrid strategy in which the top 10 candidates are first selected by MMD and the most inconsistent sample is then chosen among them.
> |Strategy|RMSE|
> |--|--|
> |Random|0.0226|
> |MMD minimizing|0.0185|
> |inconsistency|0.0206|
> |MMD minimizing+inconsistency|0.0174|
>
> These results suggest that active learning can further improve performance. We consider a more rigorous theoretical analysis of sensitivity to HR sampling strategies as an important direction for future work.
> # Q4. Error reinforcement
> As presented in Appendix R, SuperMeshNet is robust to catastrophic mutual reinforcement. Each loss term incorporates both pseudo-label–based mutual supervision and ground truth-based supervision, where the latter acts to suppress error amplification caused by imperfect pseudo-labels.
>
> To further validate stability, we have carried out controlled experiments by injecting noise into pseudo-labels. When the pseudo-label error exceeds a threshold (~0.02), we have observed error amplification during training (Fig. R3). This effect becomes significantly more pronounced when training relies solely on unsupervised loss without ground truth-based supervision (i.e., no true labels) (Fig. R4). These results provide additional evidence for the stability of our framework.
> # Q5: See our response to W3

---

> > ### Author Rebuttal · Reviewer_vpK8 · 2026-04-03
> >
> > Thanks for the reply. My concerns are addressed. I am happy to improve the evaluation accordingly.

---

> > > ### Author Response · Authors · 2026-04-07
> > >
> > > Dear reviewer vpK8,
> > >
> > > We greatly appreciate your re-evaluation. We are glad that our responses have helped clarify the key aspects of our work and have fully addressed your concerns. Thank you again for your time and consideration.

---

### Decision · Program_Chairs · 2026-04-30

**Decision:**

Accept (regular)

**Comment:**

All reviewers support acceptance and marked their concerns as fully resolved. They appreciated the simplicity and effectiveness of the centering operations and the relevance of reducing high-res data requirements. In addition, the authors provided a thorough rebuttal, including large-scale experiments at 2M nodes, physics-based metrics and an active learning strategy. I recommend acceptance. Please do include the new results and discussions into the updated manuscript.